# Universal electrochemical quantification of active site density in transition metal nitrogen carbon electrocatalysts

Guang Li[1,4], Shu-Hu Yin[2,4] ✉, Li-Fei Ji[1], Xu-Yuan Nie[3], Ting Zhu[2], Xiao-Yang Cheng[1], Jun Xu[2], Rui Huang [1], Yan-Xia Jiang [1] ✉, Bin-Wei Zhang [3] ✉ & Shi-Gang Sun [1,3]

In-situ electrochemical nitrite reduction is an established method to quantify site density (SD) of platinum-group-metal-free catalysts for PEM fuel cells. However, its poisoning mechanism remains unclear, often yielding underestimated values. Crucially, we identify a unique configuration where single metal centers adsorb two NO molecules, which challenges conventional electrochemical quantification. To resolve this, we developed an in-situ acid-assisted nitrite poisoning method (AANPM) coupled with graphene-based attenuated total reflection Fourier transform infrared spectroscopy (graphene-based in-situ ATR-FTIR). This approach quantifies SD and elucidates active site structures in transition metal-nitrogen-carbon (MNC) electrocatalysts. By incorporating the average electron transfer number for NO electroreduction (NOR), we achieve accurate SD calculations. Validated across iron/cobalt phthalocyanine molecular catalysts and pyrolyzed FeNC/CoNC materials, this method can be used to stablish structure-activity relations.

Transition metal nitrogen carbon (MNC) catalysts have emerged as promising alternatives to platinum-group-metal (PGM) electrocatalysts for proton exchange membrane fuel cells (PEMFCs)[1–3]. For decades, the development of more active MNC catalysts has relied on empirical methods[4,5], which involve systematically varying elemental precursors and synthesis conditions, then correlating these factors with kinetic current density ($J_{kin}$) and other ORR performance metrics. While this approach initially led to notable improvements, progress in the power and durability of FeNC cathodes for PEMFCs has stagnated despite extensive international efforts[6]. To drive further breakthroughs, a more rational strategy is needed—one that deconstructs the overall activity and stability of FeNC catalysts into the distinct contributions of individual Fe-based active sites. This enables the identification of the most active and robust sites, paving the way for targeted optimization strategies[7]. A critical first step is to establish

experimental methods for quantifying the number of Fe-based catalytic sites on the catalyst surface, referred to as the site density (SD)[8,9]. The SD, combined with $J_{kin}$ and the elementary charge ($e$), can then be used to calculate the intrinsic turnover frequency (TOF) of Fe-based active sites in a given FeNC catalyst, as follows[10]:

$$J_{kin}\left[\text{A}\,\text{g}^{-1}\right] = TOF\left[\text{e}\,\text{site}^{-1}\,\text{s}^{-1}\right] \times SD\left[\text{site}\,\text{g}^{-1}\right] \times e\left[\text{C}\,\text{e}^{-1}\right]$$

TOF and SD are key parameters for assessing catalytic reactivity, providing essential guidance for the rational design of more active catalysts. Strategies to enhance activity may therefore aim either to increase the SD through optimized synthesis or to boost the intrinsic TOF of the active sites.

Theoretical and computational studies have provided insights into the chemical structure of catalytically active Fe–$N_x$ single metal

[1]State Key Laboratory of Physical Chemistry of Solid Surfaces, Engineering Research Center of Electrochemical Technologies of Ministry of Education, College of Chemistry and Chemical Engineering, Xiamen University, Xiamen, China. [2]School of Microelectronics and Integrated Circuits (Jiangsu Key Laboratory of Semi. Dev. & IC Design, Package and Test), Nantong University, Nantong, China. [3]Center of Advanced Electrochemical Energy, Institute of Advanced Interdisciplinary Studies; School of Chemistry and Chemical Engineering, Chongqing University, Chongqing, China. [4]These authors contributed equally: Guang Li, Shu-Hu Yin. ✉e-mail: shyin@ntu.edu.cn; yxjiang@xmu.edu.cn; binwei@cqu.edu.cn

sites[11]. Advanced experimental techniques, including $^{57}$Fe Mössbauer spectroscopy and high-resolution STEM-EELS microscopy, have confirmed the presence of these sites in FeNC materials[12,13]. However, because a substantial fraction of Fe−N$_x$ sites are embedded within the carbon matrix and inaccessible at the surface[14], the SD cannot be determined solely from the total Fe content, even in model FeNC materials containing only Fe−N$_x$ sites. Therefore, precise quantification of electrochemically accessible Fe−N$_x$ sites is essential.

Probe molecule adsorption-desorption methods are widely applied to quantify surface sites, yet chemical specificity and sampling completeness remain challenging. Electrochemical probe molecules such as CO[15], NO[16], CN$^-$[17], and SCN$^-$[18] have been explored for FeNC materials, yet none have provided reliable quantitative surface sites densities. An ex-situ cryogenic CO chemisorption/desorption method, performed at −80 °C, offers high specificity toward Fe sites[19]. Nonetheless, its accuracy may be compromised by incomplete ORR suppression, multi-molecule adsorption, or partial site poisoning[14]. An in-situ electrochemical NO$_2^-$/NO poisoning method (NPM) at ambient temperature was recently introduced by Kucernak[20]. The method exploits the strong specific binding of NO to Fe−N$_x$ sites, with site density estimated from the stripping charge of a nominal five-electron NO-to-NH$_3$ reduction. However, uncertainties in binding stoichiometry, electron-transfer number, and incomplete suppression of ORR activity, likely due to multiple Fe−N$_x$ site types, suggest that NO may only partially deactivate the active sites, leading to incomplete sampling.

Herein, we introduce an in-situ acid-assisted nitrite poisoning method (AANPM) integrated with graphene-based in-situ attenuated-total reflection Fourier-transformed infrared spectroscopy (gra-based in-situ ATR-FTIR) to precisely quantify SD and analyze active site architectures in FeNC/CoNC catalysts. AANPM achieves ~30% higher SD values versus prior methods for molecular catalysts. In-situ ATR-FTIR resolves NO adsorption structures, identifying tetra-coordinated (MPc) and pseudo-penta-coordinated (MPc-C) sites (M = Fe/Co; C = carbon carrier). Crucially, a unique configuration where single metal centers adsorb two NO molecules was identified, which challenges conventional electrochemical quantification. Electrochemical/ATR-FTIR integration reveals 30% increased SD with site-specific structural resolution via NO molecular probes.

## Results

### The SD calculated methods of molecular catalysts

FePc uniformly dispersed on Ketjenblack EC600 carbon (FePc-KJ), synthesized via a dissolution-adsorption method (experimental details are provided in "Methods")[21], served as the model catalyst. Structural characterization confirmed homogeneous dispersion of FePc molecules due to π−π conjugation with KJ (Supplementary Fig. 1). Subsequently, NPM was employed to investigate the SD of FePc-KJ. The experimental procedures for NPM and corresponding NOR curves are shown in Fig. 1a–b and Supplementary Fig. 2. To determine whether adsorbed species were fully reduced to stable products, three consecutive NOR cycles were recorded. The difference between the 1st and 2nd NOR curves was used to calculate the NOR charge ($Q_{NOR}$). Coincidence of the 2nd and 3rd NOR curves confirmed complete reduction of NO molecules adsorbed on Fe sites. During the 1$^{st}$ NOR negative scan of FePc-KJ by NPM, the Fe$^{3+}$/Fe$^{2+}$ reduction peak at 0.78 V remained nearly unchanged versus the initial curve[22]. Conversely, a minor reduction current emerged between 0.4 and −0.2 V. These observations indicate that the catalyst's active sites exhibit poisoning resistance during NPM, with minimal poisoning species adsorbing on Fe sites. This behavior likely stems from FePc structural degradation under acidic ORR conditions (pH = 5.2) (Supplementary Fig. 3)[23].

To elucidate NO adsorption on FePc, the NO adsorption-electroreduction method (NAEM) is presented in Fig. 1c, d ("Methods", Section A 1.3). Compared to the original curve, Fe$^{3+}$/Fe$^{2+}$ redox

peaks disappear in the 1st NOR cycle, indicating NO adsorption at Fe sites. Metal-free KJ carbon exhibits negligible NOR activity (Supplementary Fig. 4). This result demonstrates that NPM's low poisoning effect stems from unstable nitrite adsorption on FePc, yielding trace amounts of NO. Figure 1d shows a distinct reduction peak at 0.2 to −0.2 V in the 1st NOR cycle. Moreover, identical second and third cycles confirm complete reduction of adsorbed NO.

As the Fe$^{3+}$/Fe$^{2+}$ reduction is a one-electron process, the total electron transfer number ($z$) for NOR was calculated from charge differences ("Methods", Section B 2.1). To investigate pH effects on NOR, NAEM tests were conducted in 0.1 M HClO$_4$ (pH = 1), 0.1 M NaClO$_4$ (pH = 7), and 0.5 M NaAc (pH = 9). Perchlorate and acetate anions were selected for their non-poisoning characteristics[24]. Corresponding NOR curves are presented in Supplementary Figs. 5–7. Consistent with observations in NaAc buffer, Fe$^{3+}$/Fe$^{2+}$ peaks disappear across pH conditions, replaced by NOR peaks (Fig. 2a). Charge integration yields $\Delta Q$(Fe$^{3+}$/Fe$^{2+}$), $Q_{NOR}$, and $z$-values (Fig. 2b). In pH-varied solutions, Fe$^{3+}$/Fe$^{2+}$ reduction peaks disappear, replaced by NOR peaks. Calculated $z$ values: 3.11 (pH = 1), 3.04 (pH = 7), 3.24 (pH = 5.2), 5.06 (pH = 9). NOR products shift from NH$_2$OH (acidic/neutral) to NH$_3$ (alkaline), confirmed by NMR (Supplementary Fig. 8). FePc favors 5e$^-$ NH$_3$ formation at pH = 9. These results demonstrate pH-dependent behavior of the FePc-KJ catalyst during NOR reduction. Contrary to previous reports[25,26], we demonstrate that FePc favors NH$_3$ formation via a 5e$^-$ pathway at pH = 9. Selective reduction of NOR to a single product is critical for SD evaluation.

Direct NO gas utilization poses hazards; therefore, we developed an AANPM generation via acidic disproportionation of NaNO$_2$ (Fig. 3a, Supplementary Fig. 9). At pH = 2, Fe$^{3+}$/Fe$^{2+}$ peaks vanish with concomitant NOR peak emergence (Supplementary Fig. 10). Figure 3b demonstrates comparable poisoning effects in AANPM and NAEM (peak assignment: Supplementary Fig. 11). For FePc-KJ, $z = 5.02$ in AANPM aligns with NAEM results. SD calculations ("Methods", Section B 2.2) yield: NAEM: $8.43 \times 10^{19}$ sites g$^{-1}$; AANPM: $8.38 \times 10^{19}$ sites g$^{-1}$; NPM: $5.27 \times 10^{16}$ sites g$^{-1}$ ($10^3$-fold lower). Kinetic current analysis ("Methods", Section B 2.3) confirms 50% higher poisoning in NAEM/AANPM versus NPM (Fig. 3c, Supplementary Fig. 12), indicating enhanced active site identification via stable NO adsorption.

Figure 4a illustrates pseudo-penta-coordinate FePc-KJ with single NO adsorption capacity due to π-π conjugation. Agglomerated FePc$_{ag}$-KJ (synthesis details are provided in "Methods") exhibits weak conjugation, preserving tetra-coordinate FePc capable of adsorbing two NO molecules. Remarkably, FePc$_{ag}$-KJ shows $z = 10.01$ in NAEM/AANPM (Supplementary Figs. 13–15)−double the value of FePc-KJ (Fig. 4b). This $z$-value divergence likely originates from distinct NO adsorption geometries modulated by active site structures. CoPc-KJ and CoPc$_{ag}$-KJ catalysts were synthesized (synthesis details are provided in "Methods"), with structures verified (Supplementary Fig. 16). NOR profiles appear in Supplementary Figs. 17–22. Co$^{2+}$/Co$^+$ redox peaks at 0.10 V confirm single-electron transfer. Calculated $z$-values: CoPc-KJ is 5.06; CoPc$_{ag}$-KJ is 10.23. SD values: AANPM is $7.59 \times 10^{19}$ (CoPc-KJ), $3.17 \times 10^{19}$ sites g$^{-1}$ (CoPc$_{ag}$-KJ). ORR poisoning effects (Supplementary Figs. 23−25) corroborate SD trends, establishing AANPM's applicability for CoPc site characterization.

### The gra-based in-situ ATR-FTIR to identify active site configurations

Quantitative charge analysis unexpectedly yielded $z \approx 10$, exceeding the theoretical value of 5e$^-$ expected for single NO molecule reduction. This result suggests simultaneous adsorption of two NO molecules at per Fe site of FePc, each undergoing a 5e$^-$ reduction, collectively accounting for $z \approx 10$. To investigate this, gra-based in-situ ATR-FTIR ("Methods", Section C, Supplementary Figs. 26−27) was used to track NO adsorption (NO$_{ad}$) and electroreduction on MPc catalysts. In the NPM process, neither FePc-KJ nor FePc$_{ag}$-KJ exhibited detectable NO$_{ad}$

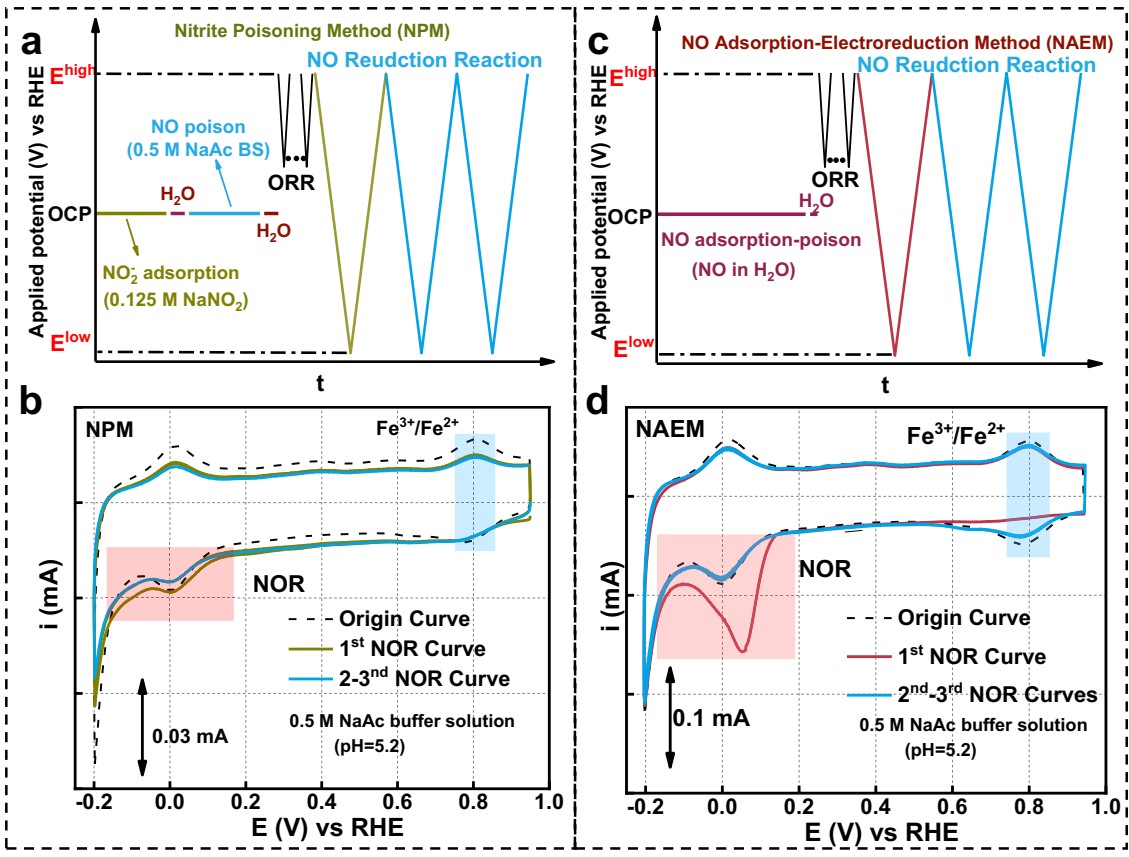

**Fig. 1 | Schematic and cyclic voltammetry curves of FePc-KJ via NOR methods. a**, **b** NPM; **c**, **d** NAEM. $E^{high}$: pre-reaction potential; $E^{low}$: post-reaction steady-state. The resistance is about 20 Ω.

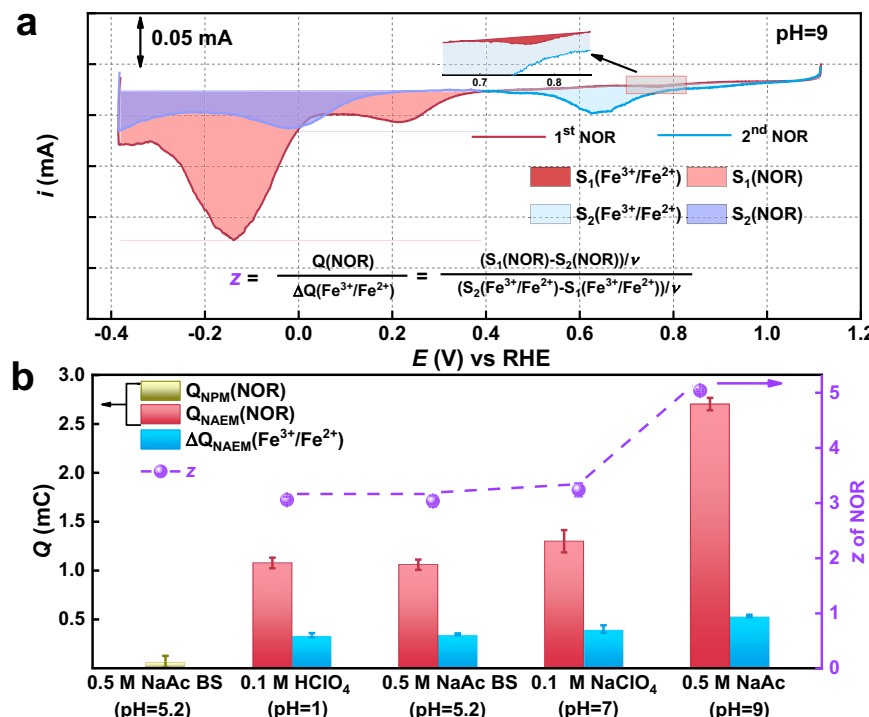

**Fig. 2 | The calculated *z* for the NOR process. a** Schematic of NOR curve integration and *z*-calculation methodology: In the first NOR curve (red), dark red shading denotes the $Fe^{3+}/Fe^{2+}$ redox peak region, while light red shading indicates the NOR reaction area. Conversely, in the second curve (blue), light blue shading represents the $Fe^{3+}/Fe^{2+}$ redox peak region, whereas dark blue shading corresponds to the NOR reaction area. **b** Integrated charge and derived *z*-values for FePc-KJ in NPM and NAEM processes.

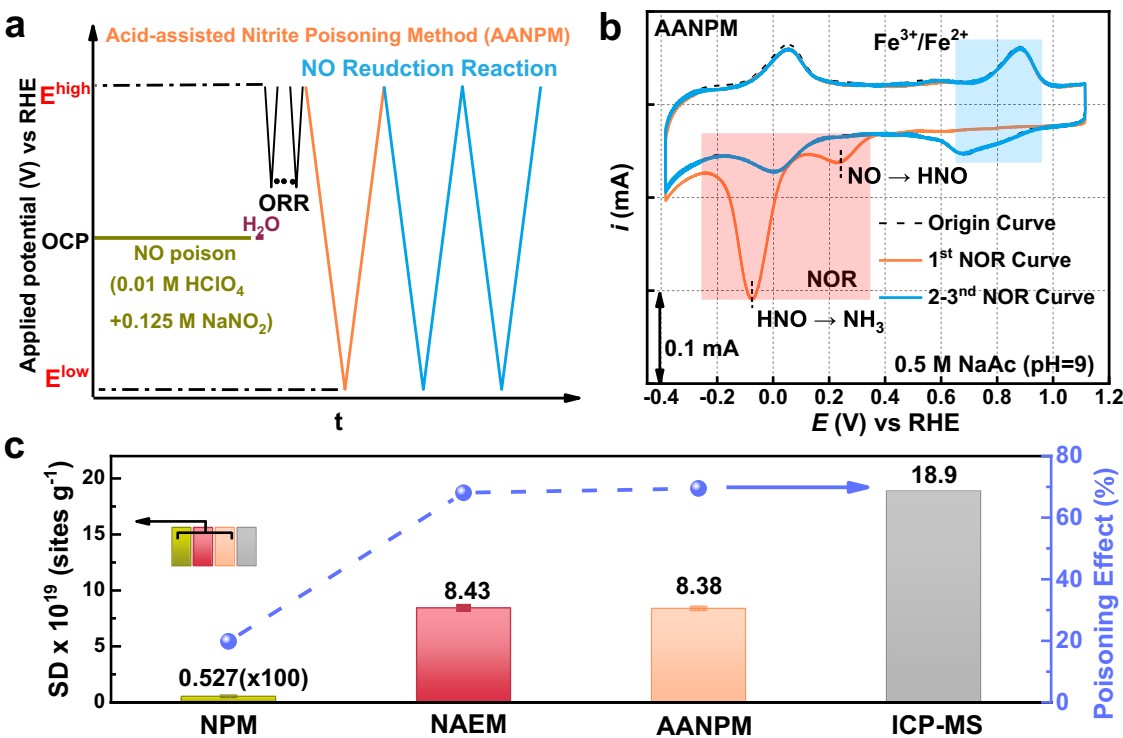

**Fig. 3 | Evaluating the active site density via AANPM and other method. a** Schematic diagram of AANPM; **b** Cyclic voltammetry curves of FePc-KJ by AANPM, the resistance is about 20 Ω; **c** The SD values and poisoning effect of ORR with different methods.

peaks after NaNO₂ treatment (Fig. 5a, b). Similarly, CoPc catalysts showed no adsorption (Fig. 5c, d), indicating negligible NO generation, corresponding with electrochemical data. In contrast, purging NO into H₂O during NAEM revealed two distinct NO$_{ad}$ peaks (-1708 cm⁻¹ and -1590 cm⁻¹ for FePc-KJ, -1704 cm⁻¹ and -1593 cm⁻¹ for FePc$_{ag}$-KJ)[27–29]. Notably, AANPM exhibited nearly identical peaks. Interestingly, CoPc catalysts also displayed dual NO$_{ad}$ peaks near 1700 cm⁻¹ and 1600 cm⁻¹ in both processes. In all cases, NO$_{ad}$ peak intensities progressively increased to a plateau over time, confirming progressive site poisoning by NO. Critically, NO adsorption saturation time in AANPM was shorter than in NAEM, presumably due to higher nitrite concentration (0.125 M vs. NO saturated solubility: 2 mM) (Supplementary Fig. 28), reflecting rapid site saturation via disproportionation-generated NO. Additionally, upward peaks at 1450–1480 cm⁻¹ in AANPM correspond to NO₂⁻[30], confirming acid-induced nitrite decomposition generating locally concentrated NO. The absence of gaseous NO peaks (2000 cm⁻¹) in both processes eliminates interference from free NO molecules. Thus, the NOR charge in NAEM and AANPM arises exclusively from NO$_{ad}$ reduction.

For NO molecules, adsorption structures differ across active sites, resulting in multiple NO$_{ad}$ peaks that indicate two distinct configurations and site types in molecular catalysts. As reported previously[27], peaks at 1700 cm⁻¹ and 1600 cm⁻¹ correspond to M(NO)₂Pc and M(NO)Pc (M = Fe, Co), signifying dual and single NO molecule adsorption at discrete active sites. Supplementary Fig. 28 further confirms M(NO)₂Pc and M(NO)Pc as dominant NO$_{ad}$ structures for FePc$_{ag}$-KJ/CoPc$_{ag}$-KJ and FePc-KJ/CoPc-KJ catalysts[16,31]. Sites with different coordination numbers preferentially adsorb one or two NO molecules to achieve stable configurations. In FePc-KJ and CoPc-KJ catalysts, π-π conjugation uniformly disperses FePc/CoPc molecules on KJ carriers, forming pseudo-penta-coordinated MPc-C. This configuration adsorbs one NO molecule, yielding stable hexa-coordinated M(NO)Pc-C (Fig. 6a). Conversely, FePc$_{ag}$-KJ and CoPc$_{ag}$-KJ retain isolated tetra-coordinated sites, adsorbing two NO molecules to form M(NO)₂Pc[32]. Density functional theory (DFT) calculations confirm that metal centers adsorb dual NO

molecules despite steric constraints (Supplementary Fig. 29). Stacked MPc crystals (M = Fe, Co) in the agglomerates further stabilize μ-(N, O) bridging structures, enabling dual NO adsorption per metal site (Supplementary Fig. 30).

The NOR pathway was investigated using in-situ electrochemical ATR-FTIR. As shown in Fig. 6b–e, negatively shifted potentials gradually restored the characteristic M(NO)₂Pc and M(NO)Pc peaks. Simultaneously, new peaks emerged at 1635 cm⁻¹ (N–H stretch) and 1438 cm⁻¹ (H–N–H bend), corresponding to NH₃ formation[32]. These observations highlight the need to account for site-specific NO adsorption when calculating z. Thus, site-averaged NO adsorption numbers ($\bar{a}$) were derived from M(NO)Pc/M(NO)₂Pc band intensity ratios (Fig. 6f). Validation was provided by the close agreement between IR-derived stoichiometries ($\bar{a} = 1.05$ for FePc-KJ, $\bar{a} = 1.85$ for FePc$_{ag}$-KJ) and those obtained from synthetic standards (Supplementary Fig. 31), with further quantitative consistency in independent in-situ electrochemical NO quantification experiments. The close agreement between infrared quantification and NOR charge-transfer measurements confirms the reliability of AANPM and in-situ ATR-FTIR for SD quantification and active-site characterization. Accordingly, the standard SD calculation was modified to integrate: a) NOR charge ($Q_{NOR}$) from AANPM under saturated NO adsorption; b) site-averaged electron transfer number ($\bar{n}$) quantified via product concentration ratios; c) NO$_{ad}$ number per site ($\bar{a}$) to calculate the total reaction electrons z (Fig. 6g).

## Application of AANPM in pyrolytic MNC catalysts

AANPM and in-situ ATR-FTIR methods were employed to determine SD and characterize active sites in pyrolysis-derived FeNC and CoNC catalysts. FeNC and CoNC were synthesized following established protocols[33,34], and structural characterization confirmed M-N₄ coordination in both catalysts, analogous to MPc model catalysts (Supplementary Fig. 32, Table 2). SD calculations were determined using multiple methods, with NOR curves presented in Fig. 7a–b and Supplementary Figs. 33–39. In NPM, the initial NOR curve exhibited a

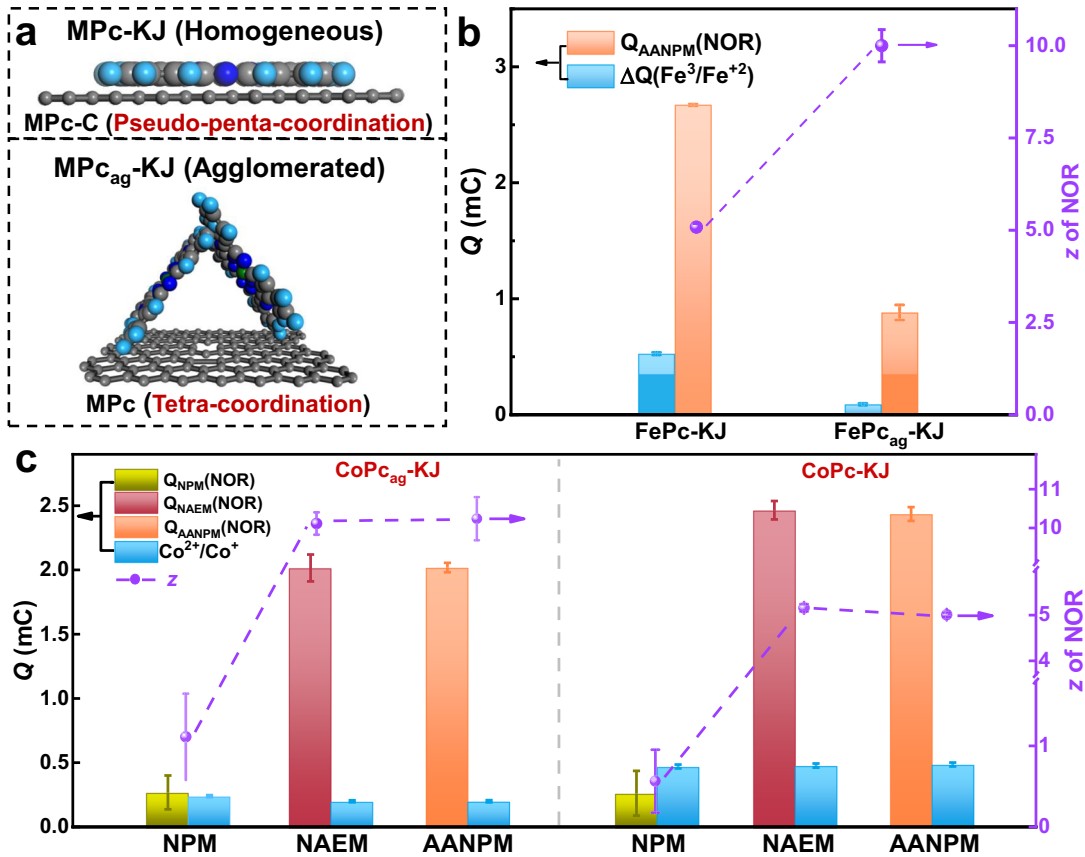

**Fig. 4 | The comparative evaluation of three active site density determination method for MPc catalysts. a** Structure schematic diagram of MPc-KJ and MPc$_{ag}$-KJ catalysts, gray sphere, dark blue sphere and light blue sphere represent C, N, H atoms, respectively; **b** The calculated integrated charge and $z$ for FePc-KJ and FePc$_{ag}$-KJ catalysts; **c** The integrated charge and $z$ of CoPc$_{ag}$-KJ and CoPc-KJ catalysts in different methods.

reduction current indicative of stable nitrite adsorption and NO production at FeNC active sites, potentially leading to site poisoning. This behavior likely stems from active site structural variations. Notably, misalignment between the first and second NOR curves further suggested incomplete NO reduction to stable products. In contrast, both NAEM and AANPM produced overlapping NOR curves near −0.62 V, indicating efficient electroreduction. For CoNC, NPM required a more negative potential (−0.38 V) to achieve complete NO reduction, whereas NAEM and AANPM both reduced NO near −0.59 V. As summarized in Fig. 7c, FeNC exhibited $Q_{NOR}$ values of 2.46 mC (NPM), 6.52 mC (NAEM), and 6.44 mC (AANPM). CoNC displayed comparable charges for NAEM (5.64 mC) and AANPM (5.93 mC), both exceeding the NPM value (3.57 mC). The higher $Q_{(NOR)}$ in AANPM compared to NPM likely reflects additional NO generation and associated active-site poisoning. ORR measurements supported this interpretation (Supplementary Figs. 40–43), revealing ~20% greater poisoning in AANPM than in NPM. The $\bar{n}$ was determined from 1-hour constant-potential NOR product analysis (Supplementary Fig. 44). NMR spectroscopy and standard curves quantified NH$_3$ and NH$_2$OH concentrations (Fig. 7d, Supplementary Fig. 45). In 0.5 M NaAc, both catalysts yielded exclusively NH$_3$, consistent with model catalysts. However, in 0.5 M NaAc BS, CoNC produced distinct NH$_2$OH peaks, resembling the behavior of FePc-KJ. For reference, $\bar{n}$ for FeNC in NPM was set to 5.

Gra-based in-situ ATR-FTIR was used to quantify the number of adsorbed NO per site ($\bar{a}$; Supplementary Fig. 46). In NPM, NO$_{ad}$ peaks at 1710 cm$^{-1}$ (FeNC) and 1705 cm$^{-1}$ (CoNC) indicated stable nitrite adsorption sites. Under AANPM, these peaks intensified and shifted to 1725 cm$^{-1}$, signifying additional NO adsorption. At low active-site densities, water interference at 1600 cm$^{-1}$ was observed, but isotope substitution experiments (D$_2$O/D$_2$SO$_4$; Supplementary Fig. 47) eliminated

this interference, revealing no peaks at 3600 cm$^{-1}$ and confirming D-O stretching at 2480 cm$^{-1}$. Stable NO$_{ad}$ spectra (Fig. 7e) assigned the 1727/1643 cm$^{-1}$ peaks to Fe(NO)$_2$NC/Fe(NO)NC and 1717/1605 cm$^{-1}$ to Co(NO)$_2$NC/Co(NO)NC. Peak integration yielded $\bar{a} \approx 1.9$ (FeNC) and 1.6 (CoNC), approaching 2 (Supplementary Fig. 48), consistent with bis-NO adsorption on tetra-coordinated M-N$_4$ sites. While penta-coordinate centers (e.g., MPc-C) can only accommodate a single NO molecule, $\bar{a} \approx 2$ was applied for SD calculations involving tetra-coordinated MNC sites. Corrected SD values (Fig. 7f) derived from $Q_{NOR}$, $\bar{n}$, and $\bar{a}$ were $2.76 \times 10^{19}$ and $2.83 \times 10^{19}$ sites g$^{-1}$ for FeNC and CoNC in AANPM, representing 33% and 20% improvements over NPM (Default $\bar{a} = 1$). Notably, ICP-MS-derived SD values ($1.56 \times 10^{20}$ and $2.07 \times 10^{20}$ sites g$^{-1}$) exceeded AANPM values by ~88% and ~86%, confirming that most high-temperature-pyrolysis-generated sites remain electrochemically inaccessible, with AANPM detecting only electroactive sites.

## Discussion

In summary, the convenient in-situ AANPM electrochemical method and gra-based in-situ ATR-FTIR were developed to accurately quantify SD and identify the active-sites structure in both molecular catalysts and MNC catalysts. Compared to conventional NPM, AANPM generates higher NO concentrations, poisoning additional active sites and thereby enabling more accurate SD quantification. Furthermore, gra-based in-situ ATR-FTIR spectroscopy identified characteristic NO$_{ad}$ peaks, revealing multiple active site configurations in MPc molecular catalysts, including tetrahedrally coordinated and pseudo-pentacoordinated sites. Notably, this finding provides the key evidence that tetrahedrally coordinated sites in MNC catalysts can adsorb two NO molecules simultaneously. This discovery has significant implications for

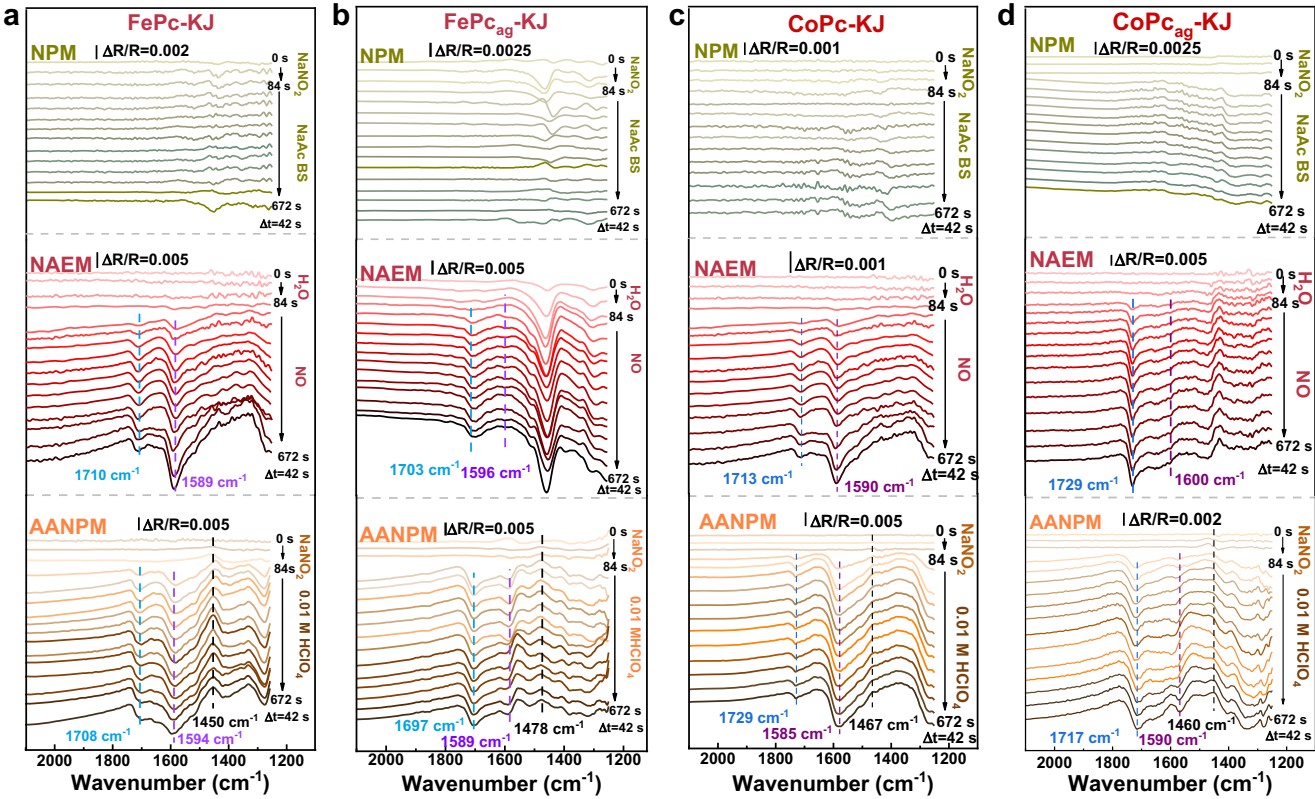

**Fig. 5 | Gra-based in situ ATR-FTIR analysis of NO adsorption modes in NPM, NAEM and AANPM processes.** Spectra are shown for **a** FePc-KJ, **b** FePc$_{ag}$-KJ, **c** CoPc-KJ, and **d** CoPc$_{ag}$-KJ, respectively. The top, middle, and bottom panels correspond to the NPM, NAEM, and AANPM processes, respectively.

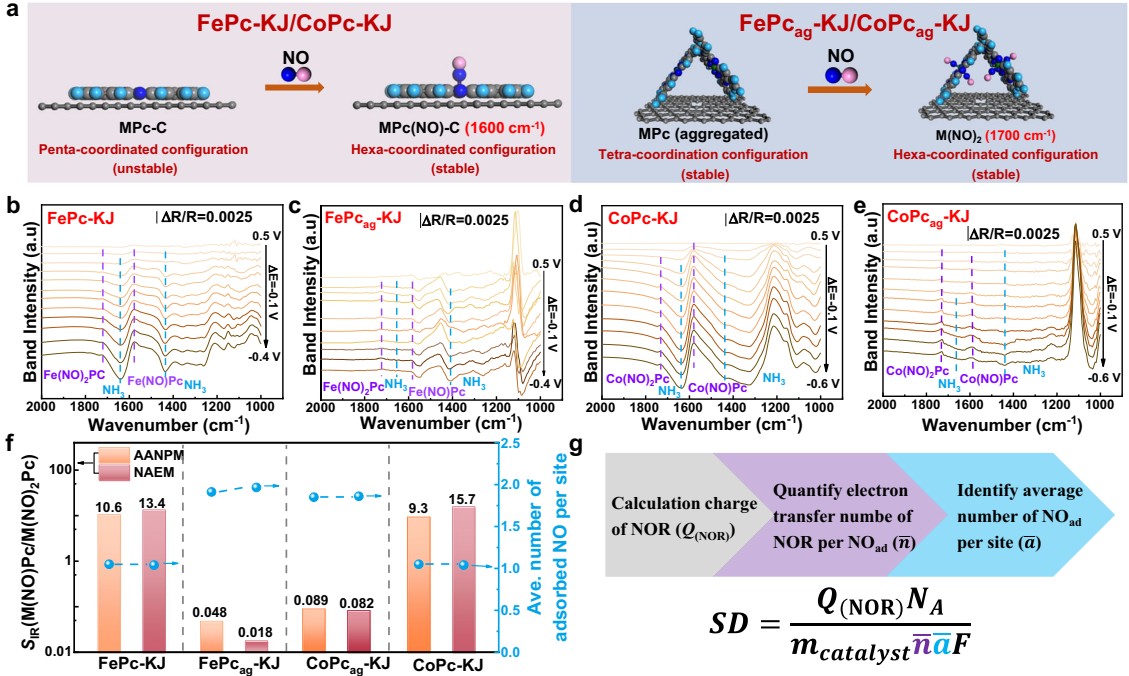

$$SD = \frac{Q_{(NOR)}N_A}{m_{catalyst}\,\overline{n}\,\overline{a}\,F}$$

**Fig. 6 | Structure analysis of active sites and correction of SD formular.**
**a** Structure schematic diagram of NO adsorption configurations of different active sites, gray sphere, dark blue sphere, light blue sphere and pink sphere represent C, N, H, O atoms, respectively; **b**–**e** NO electro-reduction spectra of different molecular catalysts; low-layer graphene transfer process; **f** integral intensity of different NO adsorption configurations and calculation theoretical $n$; **g** corrected SD formular.

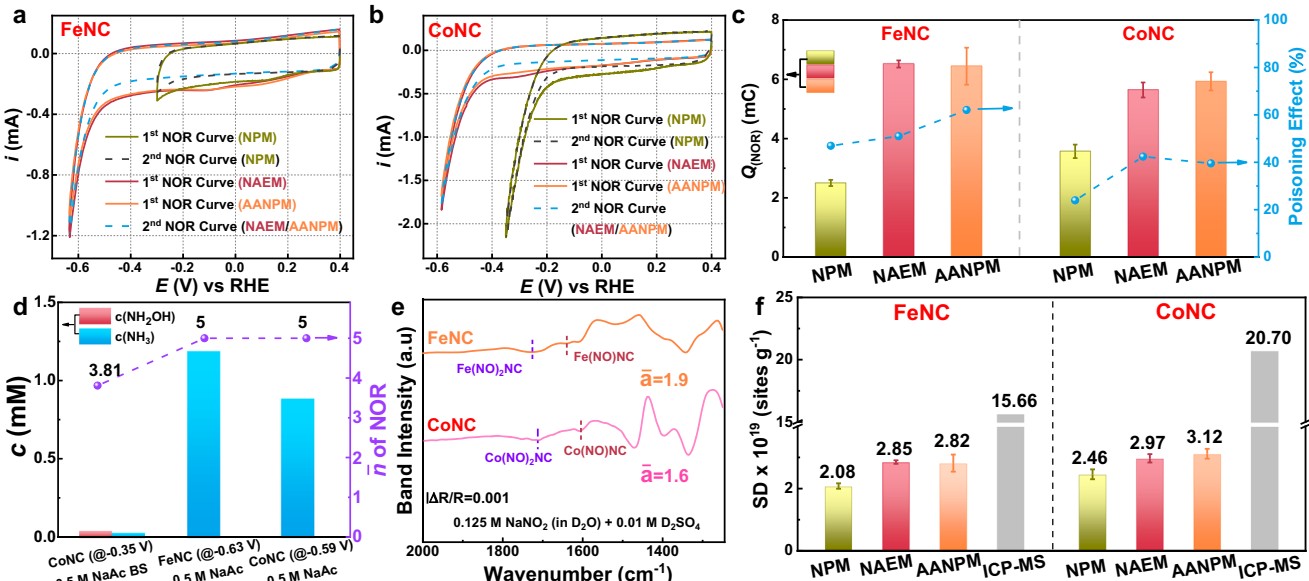

**Fig. 7 | Structure analysis and calculated SD of FeNC and CoNC. a** NOR curves of FeNC with different methods; **b** NOR curves of CoNC with different methods. Electrolyte: 0.5 M NaAc BS (pH = 5.2) for NPM, 0.5 M NaAc (pH = 9) for NAEM and AANPM; catalyst loading: 0.6 mg cm$^{-2}$. The resistance is about 20 Ω. **c** integral charge and poison effect of FeNC and CoNC; **d** product concentration of NOR and calculation of $\bar{n}$ on FeNC and CoNC in different solutions; **e** $NO_{ad}$ spectra by AANPM and calculated $\bar{a}$ in the 0.125 M $NaNO_2$ (dissolved in $D_2O$) + 0.01 M $D_2SO_4$ solution; **f** calculated SD of FeNC and CoNC with different methods, among them, $\bar{a} = 1$ for NPM and $\bar{a} = 1.9$ (FeNC) or 1.6 (CoNC) for NAEM/AANPM.

electrochemical ammonia synthesis, where NO is a key reaction intermediate whose adsorption configuration governs both reaction mechanism and kinetics. Critically, by integrating AANPM data with infrared spectral analysis, we refined the SD calculation formula through precise quantification of $Q_{NOR}$, average electrons transferred per site ($\bar{n}$) and average $NO_{ad}$ number per site ($\bar{a}$). The resulting SD values for FeNC and CoNC catalysts were ~30% higher than those obtained from conventional NPM, showing improved alignment with operationally relevant conditions. This integrated approach provides a robust and accessible strategy for SD quantification in MNC catalysts, offering a powerful tool to accelerate the rational design and development of high-performance electrocatalysts.

## Methods

### Materials and reagent

$CH_3COONa \cdot (H_2O)_3$ (99%, AR), $NaNO_2$ (99%, AR), $HClO_4$ (70% aqueous solution, ACS), $H_2SO_4$ (99%, AR), FePc (99%, AR), CoPc (99%, AR), $Zn(NO_3)_2 \cdot 6H_2O$ (99%, AR), $Co(NO_3)_2 \cdot 6H_2O$ (99%, AR), $K_2S_2O_8$ (99%, AR), PMMA (Mw = 400,000, 99%), $C_7H_6O$ (99%, GC), $C_2H_5OH$ (99%, AR), $C_3H_6O$ (99% AR) were purchased from Aladdin and used without purification. Few-layer graphene on Cu substrate (1–3 layer) was purchased from JuNa Nanomaterials (Taizhou, China). Ar (99.99%) and NO ($V_{NO}$: $V_{Ar}$ = 1:10) were purchased from Linde Industrial Gas. Deionized water used in electrochemical studies and device cleaning was from a Millipore Milli-Q system (resistivity 18.2 MΩ cm).

### Treatment of KJ-600 carbon carrier

In a typical procedure, KJ-600 (500 mg) was dispersed in 50 mL of 0.5 M $H_2SO_4$ solution. The mixture was continuous dispersed by ultrasound for 2 h to remove any remaining metal particles in KJ-600. After centrifuging to remove the acid solution, the treated KJ-600 carbon carrier was centrifuged and cleaned with ultrapure water until the pH of the centrifuged supernatant is closed to 7.

### Synthesis of FePc-KJ molecular catalyst

In a typical procedure, FePc (10 mg) was dissolved by ultrasound for 5 min in 50 mL $CHCl_3$. Then, acid-treated KJ-600 (50 mg) was added the into above solution and dispersed by ultrasound for 10 min. The mixture was ultrasound for 1 h at room temperature and stand for 4 h. The black precipitates were centrifuged and washed with $CH_3Cl$ three times and dried in vacuum at 60 °C for overnight.

### Synthesis of FePc$_{ag}$-KJ molecular catalyst

In a typical procedure, FePc (10 mg) and acid-treated KJ-600 (50 mg) were added the into above solution and dispersed by ultrasound for 10 min. The mixture was ultrasound for 1 h at room temperature and stand for 4 h. The black precipitates were centrifuged and washed with $CH_3Cl$ three times and dried in vacuum at 60 °C for overnight.

### Synthesis of CoPc$_{ag}$-KJ molecular catalyst

In a typical procedure, CoPc (10 mg) was dissolved by ultrasound for 5 min in 50 mL $CHCl_3$. Then, acid-treated KJ-600 (50 mg) was added the into above solution and dispersed by ultrasound for 10 minutes. The mixture was ultrasound for 1 h at room temperature and stand for 4 h. The black precipitates were centrifuged and washed with $CH_3Cl$ three times and dried in vacuum at 60 °C for overnight.

### Synthesis of CoPc-KJ molecular catalyst

In a typical procedure, CoPc (10 mg) was dissolved by ultrasound for 5 minutes in 40 mL $H_2SO_4$. Then, acid-treated KJ-600 (50 mg) was added the into above solution and dispersed by ultrasound for 10 min. The mixture was ultrasound for 1 h at room temperature and stand for 4 h. The mixture was slowly poured into 1.2 L frozen ultrapure water and the solution was continuous stirring to release heat from sulfuric acid dissolved in water. The cooled solution was centrifuged and cleaned with ultrapure water until the pH of the centrifuged supernatant is closed to 7. The black precipitates were dried in vacuum at 60 °C for overnight.

### Synthesis of FeNC catalyst

In a typical procedure[33], $Zn(NO_3)_2 \cdot 6H_2O$ (10 mmol, 2.975 g) and 2-methylimidazole (2-mIm, 80 mmol, 6.568 g) were separately

dissolved in 100 mL methanol by ultrasonication for 5 min. Subsequently, the Zn(NO$_3$)$_2$ solution was rapidly poured into the 2-mIm solution. The mixture was vigorously stirred for 16 h at room temperature. The resulting white ZIF-8 precipitate was centrifuged, washed with methanol three times, and dried under vacuum at 60 °C overnight. Next, 1.0 g of ZIF-8 and 1,10-phenanthroline were dispersed in a mixed solvent of ethanol and deionized water (2:1 v/v). The mixture was magnetically stirred for 12 h at room temperature and then evaporated in an oil bath at 80 °C. The dry powders were thoroughly ground, pyrolyzed under Ar atmosphere at 1000 °C with a heating rate of 5 °C·min$^{-1}$ for 1 h, and then allowed to cool naturally to room temperature. The as-prepared black products were denoted as NC. Then, 80 mg of NC and 5 mg of FeCl$_2$·4H$_2$O were uniformly dispersed by grinding in a mortar. The powders were thoroughly ground and heated to 900 °C at a ramping rate of 5 °C·min$^{-1}$ for 1 h under Ar flow. Finally, the obtained black products were directly used without further treatment and labeled as FeNC.

## Synthesis of CoNC catalyst

In a typical procedure, Zn(NO$_3$)$_2$·6H$_2$O (4 mmol, 1.19 g), Co(NO$_3$)$_2$ 6H$_2$O (1 mmol, 0.291 g) and 2-mIm (15 mmol, 1.23 g) were dissolved in 60 mL of CH$_3$OH solution and stirred until completely dissolved. The resulting mixed solution was further stirred at room temperature for 12 h. The resulting crystalline product was centrifuged to obtain a precipitate, washed five times with CH$_3$OH, and dried in a vacuum oven at 70 °C overnight. The powder was thoroughly ground and heated up 1173 K with a ramping rate of 5 K·min$^{-1}$ for 1 h under Ar gas.

## Characterizations

**Materials characterizations.** Infrared test was measured by Nicolet IRs 50 with KBr squash method. Scanning electron microscopy (SEM) images were measured on HITACHI-4800 operated at 15 kV. X-ray power diffraction (XRD) patterns were taken on a Rigaku IV XRD with Cu Kα radiation ($\lambda = 1.54$ Å, step size: 0.02 for; current: 30 mA; and voltage: 40 kV). Characterization of NO electroreduction products was measured by nuclear magnetic resonance technique (NMR, AVANCE III HD 500 MHz spectrometer). The metal composition of FeNC and CoNC were detected by an inductively coupled plasma optical emission spectrometer (ICP-MS, Agilent 720ES).

**Electrochemical measurements.** All electrochemical curves were measured on a CHI 760e electrochemical workstation using three-electrode system. For all electrochemical tests, NPM was evaluated in 0.5 M NaAc buffer (pH = 5.2), while NAEM and AANPM were studied in 0.5 M NaAc solution (pH = 9). Additional pH-specific electrolytes included 0.1 M HClO$_4$ (pH = 1) and 0.1 M NaClO$_4$ (pH = 7), with all solutions calibrated to their target pH using a certified pH meter and adjusted with NaOH/HClO$_4$ as necessary. Regarding electrode preparation, catalyst inks were formulated by dispersing materials in a mixed solvent of isopropanol:water (4:1 v/v) with 5 wt% Nafion added at 10 μL per mL of ink (final binder concentration: 0.05 wt%). Precise ink deposition volumes were applied based on catalyst type: 20 μL for molecular catalysts (2 mg mL$^{-1}$ ink concentration) and 25 μL for pyrolyzed MNC catalysts (6 mg mL$^{-1}$ ink concentration). The catalyst ink was carefully dropped onto the polished rotating ring disk electrode (RRDE, diameter is 5.61 mm, area is 0.2475 cm$^2$), leading to a desirable catalyst loading (0.6 mg cm$^{-2}$). The catalyst modified glassy carbon electrode, graphite rod, and a saturated calomel electrode (SCE) were used as the working electrode, counter electrode, and reference electrode, respectively. The potentials in this work were referred to

reversible hydrogen electrode (RHE) potentials by using the conversion equation.

$$E_{RHE}(V) = E_{SCE}(V) + 0.242V + 0.059*pH$$
$$E_{RHE}(V) = E_{Hg/Hg2SO4}(V) + 0.656V + 0.059*pH$$

Prior to testing, all electrodes underwent electrochemical activation via cyclic voltammetry (CV) in their respective electrolytes until stable responses were achieved, with special care taken for hydrophobic metal phthalocyanines—where ultrapure water rinsing and gentle N$_2$ purging were employed during activation to eliminate surface bubbles and prevent adhesion. All SD determinations followed rigorously repeated protocols ($n = 3$), with mean values and standard deviations reported. Detailed step-by-step methodologies—including instrument parameters (scan rates, potential windows), gas saturation procedures, temperature control (25 °C), and bubble mitigation strategies—are fully documented in Section A to enable exact replication.

## Section A. Process of NPM, NAEM and AANPM for active site density on molecular catalyst, FeNC and CoNC

### Section A1.1 The electrolyte preparation and storage

**0.5 M NaAc BS (pH = 5.2).** First, 68.1 g of sodium acetate trihydrate (NaAc·3H$_2$O) was dissolved in approximately 800 mL of deionized water. Subsequently, the solution was diluted to a final volume of 1 L. Glacial acetic acid was then added dropwise to adjust the pH to 5.2. This solution should be stored in a cool, dry place and kept away from organic reagents.

**0.5 M NaAc (pH = 9).** Similarly, 68.1 g of sodium acetate trihydrate (NaAc·3H$_2$O) was dissolved in about 800 mL of deionized water and then brought to a volume of 1 L. The pH was confirmed to be 9 using a calibrated pH meter. This solution is stable for immediate use; however, its pH may drift significantly during prolonged storage.

**0.1 M HClO$_4$ (pH = 1).** Approximately 500 mL of deionized water was placed into a 1 L volumetric flask. Then, 8.6 mL of concentrated HClO$_4$ was carefully added to the water (note: always add acid to water). The mixture was swirled continuously and allowed to cool to room temperature. Finally, deionized water was added to reach the 1 L mark, and the solution was mixed thoroughly. It must be stored in a cool, dry location, separated from organic substances.

**0.1 M NaClO$_4$ (pH = 7).** A total of 12.25 g of NaClO$_4$ was dissolved in roughly 800 mL of deionized water. The solution was then diluted to 1 L. Storage conditions include a cool, dry environment away from organic materials.

**0.125 M NaNO$_2$.** 0.8625 g of NaNO$_2$ was dissolved in 80 mL of deionized water within a 100 mL volumetric flask. The mixture was dispersed ultrasonically and then diluted to the mark with water. This solution is light-sensitive and prone to decomposition; therefore, it must be protected from light and used promptly.

**Section A1.2 The NO$_2^-$/NO poisoning method (NPM).** Firstly, prepared 0.5 M NaAc BS (pH = 5.2), H$_2$O, H$_2$O, 0.125 M NaNO$_2$ solution, respectively and the solution were continuously injected argon (Ar) to purge dissolved O$_2$. The working electrode was immersed in 0.5 M NaAc and conduct activation testing. The working electrode was transferred into H$_2$O and clean 1 min with 300 rotate per minute (rpm). Then, the working electrode was transferred into 0.125 M NaNO$_2$ at 5 min with 300 rpm to adsorb NO$_2^-$. The working electrode was transferred into H$_2$O at 1 min with 300 rpm to clean unstable adsorbed NO$_2^-$. The working electrode was transferred into 0.5 M NaAc BS at 5 min with

300 rpm. The adsorbed $NO_2^-$ generated NO by disproportionated reaction in acid solution, which adsorbed on active sites of catalyst. The working electrode was transferred into $H_2O$ at 1 min with 300 rpm to clean unstable adsorbed NO and transferred into newly 0.5 M NaAc BS. After activation, the NO reduction reaction (NOR) was conducted and recorded three cycles data with different potential range for different materials. The firstly cyclic voltammetry was NOR curve and the other cyclic voltammetry was overlapping to ensure that adsorbed NO was completely reacted. For ORR test after NO adsorption and poison method, it maybe adsorbed unreactive oxygen. The working electrode was set at 0.35 V-0.55 V (*vs.* RHE) to react oxygen poisoned process at 300 rpm before NOR. Meanwhile, the NOR curves were recorded for three cyclic curves at 300 rpm.

**Section A1.3 The NO adsorption and electroreduction method (NAEM).** Firstly, prepared 0.5 M NaAc solution (pH = 9), $H_2O$ solution, respectively and the solution were continuously injected Ar to purge dissolved $O_2$. The working electrode was immersed in 0.5 M NaAc and conduct activation testing. Then, the working electrode was transferred into $H_2O$ and clean 1 min with 300 rpm. After, the working electrode was transferred into Ar purged $H_2O$ solution and continuously injected NO ($V_{NO}$: $V_{Ar}$ = 1:10). The working electrode adsorbed NO at 10 min with 300 rpm. Moreover, the working electrode was transferred into $H_2O$ at 30 s with 300 rpm to clean unstable adsorbed NO. The NOR was conducted and recorded three cycles data with different potential range for different materials. The firstly cyclic voltammetry was NOR curve and the other cyclic voltammetry was overlapping to ensure that adsorbed NO was completely reacted. For ORR test after NO adsorption and poison method, it maybe adsorbed unreactive oxygen. The working electrode was set at 0.35 V-0.55 V (*vs.* RHE) to react oxygen poisoned process at 300 rpm before NOR. Meanwhile, the NOR curves were recorded for three cyclic curves at 300 rpm.

**Section A1.4 The acid-assisted nitrate ($NO_2^-$) poisoning method (AANPM).** Firstly, prepared 0.5 M NaAc solution (pH = 9), $H_2O$, 0.125 M $NaNO_2$ and 0.1 M $HClO_4$ solution, respectively and the solution were continuously injected Ar to purge dissolved $O_2$. The working electrode was immersed in 0.5 M NaAc and conduct activation testing. The working electrode was transferred into $H_2O$ and clean 1 min with 300 rpm. Then, the working electrode was transferred into Ar purging 18 mL 0.125 M $NaNO_2$ solution and added 2 mL 0.1 M $HClO_4$ solution (pH = 1). The acid solution would enhance disproportionated reaction and generated enough NO to poison active sites. The working electrode was immersed in acid solution at 10 min with 300 rpm. Moreover, the working electrode was transferred into $H_2O$ at 30 s with 300 rpm to clean unstable adsorbed NO. The NOR was conducted and recorded three cycles data with different potential range for different materials. The firstly cyclic voltammetry was NOR curve and the other cyclic voltammetry was overlapping to ensure that adsorbed NO was completely reacted. For ORR test after NO adsorption and poison method, it maybe adsorbed unreactive oxygen. The working electrode was set at 0.35 V-0.55 V (*vs.* RHE) to react oxygen poisoned process at 300 rpm before NOR. Meanwhile, the NOR curves were recorded for three cyclic curves at 300 rpm.

**Section B. Calculation process of the parameters**
**Section B2.1 The Calculation of total electron transfer number (*z*) for NOR on the MPc molecular catalysts.** The electroreduction electricity and total electron transfer number (*z*) for NOR were

calculated by the following Eqs. (1)–(8):

$$S(NOR) = S_1(NOR) - S_2(NOR) \tag{1}$$

$$Q(NOR) = \frac{S(NOR)}{\upsilon} \tag{2}$$

$$S_{1e}(Fe) = S_2\left(Fe^{3+}/Fe^{2+}\right) - S_1\left(Fe^{3+}/Fe^{2+}\right) \tag{3}$$

$$\Delta Q\left(Fe^{3+}/Fe^{2+}\right) = \frac{S_{1e}(Fe)}{\upsilon} \tag{4}$$

$$S_{1e}(Co) = S_2\left(Co^{2+}/Co^+\right) \tag{5}$$

$$\Delta Q\left(Co^{2+}/Co^+\right) = \frac{S_2\left(Co^{2+}/Co^+\right)}{\upsilon} \tag{6}$$

$$z(Fe) = \frac{Q_{Fe}(NOR)}{\Delta Q\left(Fe^{3+}/Fe^{2+}\right)} \tag{7}$$

$$z(Co) = \frac{Q_{Co}(NOR)}{\Delta Q\left(Co^{2+}/Co^+\right)} \tag{8}$$

where $S_1(NOR)$ and $S_2(NOR)$ are integral area of the first and second NO reduction curve. $S(NOR)$, $Q(NOR)$ and $\upsilon$ are NOR integral area, electricity of NOR integral area and sweep speed of cyclic voltammetry. $S_2\left(Fe^{3+}/Fe^{2+}\right)$ and $S_1\left(Fe^{3+}/Fe^{2+}\right)$ are integral area of the reduction peak of $Fe^{III}/Fe^{II}$ in the first and second NO reduction curve. $Q_{1e}(Fe)$ and $S_{1e}(Fe)$ are the electricity of the reduction peak of $Fe^{3+}/Fe^{2+}$ and integral area of the reduction peak of $Fe^{3+}/Fe^{2+}$. $S_{1e}(Co)$, $S_2\left(Co^{2+}/Co^+\right)$ and $Q_{1e}(Co)$ are integral area of the reduction peak of $Co^{2+}/Co^+$, integral area of the reduction peak of $Co^{2+}/Co^+$ in the second NO reduction curve and the electricity of the reduction peak of $Fe^{3+}/Fe^{2+}$. $z(Fe)$ and $z(Co)$ are the number of electron transfer of NOR.

**Section B2.2 Quantification of active centers.** The SD was obtained according to the previously reported nitrite reduction method by Kucernak et al. The different poisoned methods and electricity of NOR were measurement in the last section and calculated by the following Eq. (9). Meanwhile, the total active sites were calculated based on the result of ICP-MS by the following Eq. (10):

$$SD\left(site\,g^{-1}\right) = \frac{Q(NOR)*N_A}{V_{ink}C_{ink}nF} \tag{9}$$

$$SD(ICP-MS) = \frac{Fe(wt\%)*N_A}{56}*10^6 sites\,g^{-1} \tag{10}$$

where $Q(NOR)$ is the electricity of NOR. $V_{ink}$, $C_{ink}$ are volume and concentration of loaded catalyst. $n$, $F$, $N_A$ are number of transferred electrons, Faraday's constant (96485 C mol$^{-1}$) and Avogadro constant (6.02×10$^{23}$ sites mol$^{-1}$). $Fe(wt\%)$ is the atomic mass fraction, measured by ICP-MS.

**Section B2.3 The poisoning effect in NPM, NAEM and AANPM processes.** Oxygen reduction reaction (ORR) tests were performed in 0.5 M NaAc BS (pH = 5.2) and 0.5 M NaAc solution (pH = 9). For avoiding the effect of NOR reduction, RRDE measurements were

conducted by linear sweep voltammetry (LSV) with the potential range from 0.3 to 1.1 V *vs.* RHE at 900 rpm with a scan rate of 10 mV s$^{-1}$. The origin curve, poison curve and recovered curve were recorded at $O_2$-saturated solution in origin state, poisoned state and recover state after NOR at 900 rpm. The kinetic current densities ($j_k$) involved during the ORR process were determined by analyzing Koutecky-Levich (K-L) Eq. (11):

$$\frac{1}{i} = \frac{1}{i_k} + \frac{1}{i_L} \tag{11}$$

where $i$ is the measured current density, $i_L$ and $i_k$ are the limiting and kinetic current densities. The poisoned effect of different methods was following Eq. (12):

$$\text{Poisoning effect} = \frac{\Delta j_k}{j_k^{Origin}} = \frac{j_k^{Origin} - j_k^{Poisoned}}{j_k^{Origin}} \tag{12}$$

where $j_k^{Origin}$, $j_k^{Poisoned}$ are kinetic current densities of origin curve and poisoned curve.

**Section B2.4 Quantitative analysis the product of NO reduction reaction and the average electron transfer number of NOR on single active sites ($\bar{n}$) of FeNC and CoNC.** For more accurately quantitative and analysis of product of NO reduction reaction, the nuclear magnetic resonance was used detected and quantitatively calculated reaction production. H-type electrolytic cell was used to collect product. The working electrode and reference electrode were one side. The count electrode was other side. For avoiding the effect of Cl$^-$, the reference electrode was Hg/Hg$_2$SO$_4$ electrode (E$_{Hg/Hg2SO4}$ = 0.656 V). The working electrode was immersed in 45 mL 0.5 M NaAc and 0.5 M NaAc BS solution (pH = 5.2) to collect different reaction products, respectively. The solution was continuously injected Ar to purge dissolved $O_2$. Then, NO was continuously injected for 10 min to saturate the solution and electroreduction for 1 h. After electrolysis, Ar was injected into to purge NO. Taking 15 mL original reaction solution and added 1 mL formaldehyde solution (HCHO:H$_2$O = 1:1). The mixture solution was sonicated 10 mins for hydroxylamine completely reacting with formaldehyde. The test solution of NH$_2$OH: 450 μL mixture solution +100 μL D6-DMSO + 50 μL D6-DMSO of maleic acid (4.8 mg mL$^{-1}$). The test solution of NH$_3$: 450 μL original solution +100 μL D6-DMSO + 50 μL D6-DMSO of maleic acid (4.8 mg mL$^{-1}$) + 20 μL concentrated sulfuric acid. By using the standard curve method, the concentration of NH$_3$ and NH$_2$OH was calculated. And the average NET of NOR on single active sites ($\bar{n}$) of FeNC and CoNC was calculated following Eq. (13):

$$\bar{n} = \frac{c(NH_2OH) * 3}{c(NH_2OH) + c(NH_3)} + \frac{c(NH_3) * 5}{c(NH_2OH) + c(NH_3)} \tag{13}$$

where $c(NH_2OH)$, $c(NH_3)$ are the concentration of NH$_2$OH and NH$_3$ in the solution.

**Section C. Preparation and measurement of gra-based in-situ ATR-FTIR**

**Section C3.1 Process of graphene transfer.** The few-layer graphene on Cu substrate (F-Gr@Cu, 1.5 cm *1.5 cm) was fixed on glass. Its surface was spun coat polymethyl methacrylate (PMMA, M$_w$ = 996 K, 0.4 g mL$^{-1}$, dissolved in anisole) and dried for 12 h at 25 °C to obtain PMMA@F-Gr@Cu structure. The PMMA@F-Gr@Cu was immersed in potassium persulfate solution (0.1 g/mL) for 6 h to etch Cu substrate (PMMA@F-Gr). The PMMA@F-Gr was washed by ultrapure water for three times and transferred on the surface of Si prism (PMMA@F-Gr@Si). The PMMA@F-Gr@Si was immersed in acetone to remove

PMMA and few-layer graphene was fixed on Si prism as conductive substrate (F-Gr@Si).

**Section C3.2 In-situ ATR-FTIR measurement.** To prepare a homogeneous ink, 2 mg catalyst was dispersed in in solution containing 990 μL ethanol and 10 μL 5 wt% Nafion under ultrasound for 1 h. 40 μL ink was slowly dropped onto the surface of F-Gr@Si and dried at room temperature as working electrode. The carbon rod and saturated calomel electrode (SCE) were as counter electrode and reference electrode, respectively. Based on above process of different calculation methods for active site density, the NO adsorption process and NO electroreduction reaction was detected by in-situ ATR-FTIR.

Electrochemical in-situ ATR-FTIR was employed by a Nicolet 6700 FTIR spectrometer equipped with a liquid nitrogen-cooled MCT-A detector, covering a spectral range from 4000 to 1000 cm$^{-1}$. The 263 A workstation was used as a potentiostat for potential control. In the NO-adsorption study of NPM, NAEM and AANPM, the time-resolved spectroscopy (TRS) was employed under potential control at OCP, with 200 scans and a resolution of 8 cm$^{-1}$. Firstly, the solution was purged with Ar for 10 min to establish an inert atmosphere. Reference spectra were then recorded in the Ar atmosphere for 42 s. Subsequently, the system was switched to different situation for different method and spectra were collected for 588 s. In the NOR spectra measurements, the multi-stepped FTIR method (MS-FTIR) was used to collect spectra during the NOR test from 0.5 V to −0.4 V at 0.1 V intervals. The reference spectra were recorded in the at low potential. The spectra were calculated by the relative change in reflectivity ($\Delta R/R$) with the following Eq. (14). The $R(E_S)$ and $R(E_R)$ represent the single-beam spectrum recorded at the sample setting potential $E_S$ and the reference potential $E_R$, respectively. In all spectra, $E_R$ was calculated by 0.1 V unless emphasized.

$$\frac{\Delta R}{R} = \frac{R(E_S) - R(E_R)}{R(E_R)} \tag{14}$$

**Section C3.3 The calculation of the NO$_{ad}$ number per site ($\bar{a}$) by NO$_{ad}$ peak intensity of spectra.** For calculating the average number of NO$_{ad}$ molecular per sites ($\bar{a}$), the band intensity was integrated and calculated by following Eq. (14):

$$\bar{a} = \frac{S(M(NO)Pc) + S(M(NO)_2Pc)}{\left(S(M(NO)Pc) + \frac{S(M(NO)_2Pc)}{2}\right)} \tag{15}$$

where $S(M(NO)Pc)$ and $S(M(NO)_2Pc)$ are the band intensity of M(NO)Pc and M(NO)$_2$Pc in the spectra.

## Data availability
The data that support the findings of this study are available within the article and its Supplementary Information files. All other relevant data supporting the findings of this study are available from the corresponding authors upon request. Source Data file has been deposited in Figshare under https://doi.org/10.6084/m9.figshare.30101062.

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

## Acknowledgements

This work was granted by the National Natural Science Foundation of China (22288102 of Shi-Gang Sun, 22172134 of Yan-Xia Jiang, 22279011 of Bin-Wei Zhang and 22502091 of Shu-Hu Yin), Natural Science Foundation of Jiangsu Province, China (Grant No. BK20250958 of Shu-Hu Yin), Strategic Research and Consultancy Project of the Chinese Academy of Engineering (Grant No. 22-HN-ZD-02 of Shi-Gang Sun), Fundamental Research Funds for the Central Universities (Grant No. 2024JAIS-ZX003 of Shi-Gang Sun), Large Instruments Open Foundation of Nantong University (Grant No. KFJN2561 of Shu-Hu Yin) and the Innovation and Development Joint Fund Project of Chongqing Natural Science Foundation (Grant No. CSTB2023NSCQ-LZX0070 of Bin-Wei Zhang). We are thankful to the Beijing Synchrotron Radiation Facility (1W1B, BSRF) for help with characterizations. We also thank Meng-Wei Lin for the help in the NMR measurements.

## Author contributions

The project was conceptualized by G.L., S.Y., Y.J., and B.Z. B.Z., Y.J., and S.S. supervised the project. G.L. and S.Y. synthesized the catalysts, conducted electrochemical tests, ATR-FTIR, and performed related data processing. L.J. performed NMR analyses, while X.N. and X.C. participated in material synthesis and characterization. T.Z. and S.Y. performed theoretical calculations. R.H. provided conceptual guidance. S.Y. and G.L. wrote the manuscript, with revisions contributed by B.Z., J.X., Y.J., and S.S. All authors discussed the results and commented on the manuscript; G.L. and S.Y. contributed equally.

## Competing interests

The authors declare no competing interests.
