## [Transparent Peer Review file · Nature Communications]

Universal electrochemical quantification of active site density in transition metal nitrogen carbon electrocatalysts

Corresponding Author: Dr Bin-Wei Zhang

Version 0:

Reviewer comments:

Reviewer #1

(Remarks to the Author)

In this manuscript, the authors presented the determination of site density and active structure by an in-situ acid-assisted nitrite poisoning (AANPM) approach integrated with graphene-based ATR-FTIR. The quantified average electron transfer number of NO electroreduction was used to access the accurate site density. This approach demonstrates broad applicability across both molecular model systems of Fe/Co phthalocyanines and pyrolyzed M-N-C materials, which may provide insight into direct determination of the structure-activity relationship. The results are somewhat interesting, and could be considered further unless the following issues and concerns are addressed.

1. The authors used the AANPM method to measure the site density of M-N-C catalysts under alkaline, neutral and acidic electrolytes. The precondition is that the model catalysts maintained structural stability. Did the catalyst undergo dynamic reconfiguration in acidic or alkaline media? Does it have any effect on the poisoning process?
2. In acidic environment, HER becomes competitive with the NOR. In such a case, the generation of H₂ and other N-containing reaction products cannot be ignored. Their potential impact on electron transfer number calculations and methods of elimination need to be discussed.
3. What are the advantages and disadvantages of the new method in terms of sensitivity, operational complexity, and applicability scenarios compared to traditional methods such as CO, CN⁻, and SCN⁻ poisoning methods? A systematic comparison is needed in the discussion section.
4. The key of this AANPM method is to ensure NO molecule can be only adsorbed on the active sites of the M-N-C catalysts. How to prove that the poisoning of NO occurs only in the active site (e.g., M-N₄ sites), but not in the inactive region (e.g., carbon substrate)?
5. It is suggested to supplement the DFT calculations to simulate the adsorption configuration and electron transfer path of NO on typical M-N-C sites, so as to provide theoretical support for the spectral analysis and reaction mechanism.
6. In Figure 5b and 5d, numerous vibration peaks exist in the range of 1200-1600 cm⁻¹, and these peaks change as the reaction time increases. What adsorbed species do they represent?
7. In the method section, the in situ electrochemical cell design, ATR-FTIR parameters and data processing flow should be described in detail.

Reviewer #2

(Remarks to the Author)

Summary and Significance:

This manuscript presents a comparative study of three methods—NPM (previously published by the Kucernak group), NAEM, and a newly proposed AANPM method—for determining active site density (SD) in non-precious metal (non-PGM) catalysts. The AANPM method modifies the original NPM approach by adjusting the poisoning conditions, and the authors test its applicability across six different Fe- and Co-based catalysts. The study also employs gra-based in situ ATR-FTIR to provide structural insights into the active sites.

The development of robust and accurate methods for determining active site density in non-PGM catalysts is of substantial importance to the field of electrocatalysis, particularly for oxygen reduction reaction (ORR) studies. However, the true nature and identity of the active sites in non-PGM catalysts remain a subject of ongoing debate. As such, it is inherently challenging to conclude whether the newly proposed AANPM method provides a more accurate measurement of site density than the previously published NPM method.

Novelty and Comparison to Literature:

While the NPM method is already known, the introduction of AANPM with modified poisoning conditions appears to be a novel contribution. The study draws attention to the reproducibility and applicability of SD measurements across different metal centers (Fe, Co), which is relevant given the diversity of non-PGM catalyst structures.

Validity of Conclusions and Required Clarifications:

Although the paper presents compelling preliminary results, several key aspects need to be clarified or experimentally supported to validate the authors' claims:

1. Figure 1a/c: The meaning of E_{high} and E_{low} should be clearly defined.
2. Figure 1d: The first NOR scan lacks an oxidation curve. Is the oxidation behavior consistent across all cycles?
3. Figure 2a: The color labels seem wrong.
4. NOR at different pH levels: At pH 1 and pH 9, an additional reduction peak (~ 0.3 V in pH 1 and ~ 0.2 V in pH 9) appears. What is its origin?
5. NOR electron transfer number >5 : This suggests that NO may be affecting non-metal sites. Could this indicate an overestimation of SD due to non-selective poisoning?
6. The authors mention the use of RRDE, but key parameters such as rotation speed are missing. This information is critical for reproducibility.
7. It is unclear whether the observed SD truly reflects only metal-N_x sites. Evidence is needed to rule out NO interactions with the carbon/nitrogen framework.
8. The performance of catalysts before and after recovery (in SI Figure 8) should be better explained—why is the recovered ORR performance worse than the poisoned state?
9. The AANPM method reports similar SD for Fe-NC and Co-NC despite significant differences in metal loading and ORR activity. This calls into question the method's site specificity.

Additional Evidence and Mechanistic Insight:

- Beyond ATR-FTIR, is there further evidence (e.g., XPS, Mössbauer, or DFT modeling) to support the presence of dual NO adsorption or selective binding to metal centers?
- The two reduction peaks in Figure 3b (NOR) should be clearly assigned to specific site interactions or mechanistic steps.

Reproducibility and Data Transparency:

- The experimental section should provide comprehensive details—including electrolyte composition, electrode preparation procedures, and measurement protocols—to ensure reproducibility of the results. It is noted that all site density (SD) determination methods were repeated three times, which is commendable and supports the reproducibility of the measurements.

Reviewer #3

(Remarks to the Author)

In this work, the authors report on a study of nitrite poisoning and in situ generated NO, as molecular probes to count the number of active sites in molecular Metal-N₄ catalysts and pyrolytic M-N₄ catalysts.

The study shows that a higher and presumably more accurate number of active sites is quantified with their methods as with nitrite stripping.

Overall, the work could be of interest and might bring sufficient novelty compared to previous works on site density counting method for M-N-C catalysts if the main issues below are addressed:

- 1) the novelty of the results reported on iron phthalocyanine (Fe-Pc) needs to be clarified. Fe-Pc has been studied a lot in the past. Is the pH effect on the type of products obtained new, or was it reported before in the past? (rows 132 to 152)
- 2) the authors identify the NO or nitrite-induced reduction products only indirectly, for the molecular catalysts, i.e. from an assessed number of electrons, in turn obtained from comparative electric charges. The reduction products must be identified qualitatively at least, if not quantitatively, also directly by appropriate independent (non electrochemical).
- 3) the authors study materials with 'agglomerated phthalocyanines' and use the scheme in figure 4a - bottom to image the possible situation. What are the experimental data suggesting such an unexpected arrangement of Fe-Pc? Fe-Pc molecules would tend to aggregate by planar stacking, forming well known crystalline structures, although this seems not to have taken place as no discernible peak is seen in XRD in fig S1 of FePc-ag-KJ. The number of electrons for FePc-ag/KJ of 10 however is not supported. (which product of NO reduction could reach 10 electrons?? this likely is an artifact and perhaps some physicochemical effect leads to an apparent n of 10 (2x5) while in reality it is still 5?) - if the n of 10 implies that 2 NO adsorb on a single Fe (or Co) center, then the definition of n in present work is not truly the electron number (number of electrons exchanged per one NO) but this number n times the number of NO molecule adsorbed per metal center, and this should be clarified (also in equation) from the start of the paper, not in the middle of it. Likely, the agglomerated materials in fact contain M-Pc crystals that are stacked FePc-s parallel and this may stabilise N-NO-Fe-NO-N bridges, with 2 NO adsorbed per Fe, one above and one below each FePc molecule.
- 4) as metal-phthalocyanines typically show metal-related redox peaks, the electric charge associated with such peaks can be used to quantify the number of active sites. This approach should be used to quantify site density and compare it to the values assessed by NPM, NAEM and AANPM approaches used in present work. It implies also that counting metal-centered active sites is not an issue for metal phthalocyanines, and the real challenge is in counting such sites in pyrolysed materials, where no or weak redox is often observed, despite high ORR activity.

5) lines 349 - 351 : looking at fig 7d, the reviewer strongly wonders how the authors can reliably quantify the intensity of the two different 'peaks' due to their widths - unclear shape - and the non-flat baseline ?? this raises a huge doubt (to the least) on the accuracy of the calculated number of NO molecule adsorbed per metal center ! Moreover, the assignment of each of these bands to $\text{Co}(\text{NO})_2\text{NC}$ and $\text{Co}(\text{NO})\text{NC}$ (or Fe as metal) is not strongly supported : it is well known that pyrolysed catalysts contain multiple site structures (even if all are M-N₄, as has been shown for Fe-N-C by Mossbauer spectroscopy, most catalysts showing two doublets D1 and D1 in Mossbauer spectra). Thus, the two vibration bands may also be assigned to a single NO adsorbed on different metal -N₄ site configurations .

less critical but still important questions:

a) the site density measured by nitrite stripping for Fe-Pc/KJ is ca 1000-times lower than obtained by NAEM or AANPM (fig 3c) . While nitrite stripping has been reported to underestimate to some extent the site density in pyrolysed materials, the difference with other methods was clearly not that high . Could the authors comment on this ? is it specific to Fe-Pc or phthalocyanines in general , but not to pyrolysed materials ?

b) what is the assignment of the redox at -0.1/0.1 V seen in fig . 3b for example ? and what are the assignments of the first NOR reduction wave starting at 0.35 V and of the second reduction wave with onset at ca 0.1 V ?

c) figure 2a and the use/definition of the different electric charges is unclear . In particular the S1 ($\text{Fe}^{3+}/\text{Fe}^{2+}$) is not visible .

d) the authors assign the NOR products at acidic, neutral and high pH to NH_2OH for the two first and to NH_3 for the last (lines 147-148), based on the n number. However, the n number in fig 2 is much closer to 2 e- than 3 e- at pH 2 and 5.2. On line 151, the authors write that NH_3 is identified as a more stable product. What is the support for this ? moreover, for site density assessment, what matters most is selectivity, not stability (i.e. important is that NO is reduced into a single product, not several products)

e) on line 218-219 , the authors state here that they previously (in the paper) found that 2 NO molecules can adsorb per metal center (likely referring to the agglomerate labelled catalysts). This comes to a surprise, as this conclusion was not explicitly stated earlier in the paper, and is an important one to follow the authors's thoughts. Please clarify earlier in manuscript when this conclusion could be drawn (the n = 10 ?)

f) line 253-254 : which previous works ? please cite them . Moreover, if the assignment is correct, then the agglomerated catalysts with n= 10 (2 adsorbed NO molecule per metal center) should only show one out of the two vibration bands, but this seems not to be the case (except for FePcag/KJ and the NAEM case, fig 5b, second row)?

g) line 257 : "The NOad band intensity in the Supplementary Fig. 26..." , wrong sup fig number ? the sup fig 26 does not show that . maybe it is the sup fig 23 ? in figure caption of sup fig 23, please precise the vibration frequency of the band intensity considered. It is unclear .

h) "The self-consistency between theoretical and experimental results" : there is no 'theoretical' results in the present work ??

i) figure 7 : the experimental conditions are not given in the figure caption ! electrolyte, pH, catalyst loading, etc. Moreover, the polarisation down to -0.6 V vs RHE of FeNC or CoNC is not desirable as it can lead to the formation of metallic nanoparticles from M-N₄ sites. Is it really a RHE scale, or SHE scale ? (it is surprising to see no hydrogen evolution before -0.5 V vs RHE)

j) line 312 : what is meant with misalignment ? the reviewer could not see such misalignment or differences between 1st and 2nd NOR curve between NPM and the other two approaches, in Figure 7.

k) sup fig 34 e-f : what is NH_2OH and too-small-to-be-seen spectrum in red in bottom ,and same question for NH_4^+ in fig . f ... ? please add on same scale the NMR spectra for NH_2OH and NH_4^+ in the same scale as the others, to make this visible and understandable .

l) quoted : line 346 " The stable NOad spectra are shown in the Fig. 7e. " Fig 7e does not show spectra ??

Version 1:

Reviewer comments:

Reviewer #1

(Remarks to the Author)

In the revised manuscript, the authors have supplemented comprehensive experiments and calculations to address the issues and concerns raised by the reviewers. I am satisfied with the changes in the revised manuscript, thus this manuscript could be considered for publication in Nature Communications at the present stage.

Reviewer #2

(Remarks to the Author)

In general, the authors responded to the comments very well and provided detailed explanations. I have just one final comment. The effective buffering range of acetate is approximately pH 3.6–5.6. Therefore, preparing an electrolyte at pH 9 with sodium acetate would not function as a proper buffer. If the local pH changes under these conditions, the potential versus RHE could be reported incorrectly.

Reviewer #3

(Remarks to the Author)

The authors clarified and strengthened the discussion and conclusions reported in the original manuscript.

Beside one minor issue (see below), the revised manuscript is recommended for publication

issue : in the revised version, the authors assign the redox peak at low potential seen for FePC to FeII/FeI. While this might sound logical, it has been shown that for FePc, the low potential redox in aqueous media is due to a reduction/oxidation of the Pc ligand, rather than of the Fe atom (<https://pubs.acs.org/doi/full/10.1021/acs.jpcclett.7b01126>) .

Responses to Reviewers' Comments

We sincerely appreciate the reviewers for the expert and valued comments and constructive suggestions which greatly help us to improve the quality of this submission. In the following, we carefully respond the reviewers' comments point by point (texts in blue), and describe the corresponding revisions made on the manuscript and Supplementary Information (orange font). The figures in this Response Letter are named as Figure R**.

Reviewer #1 (Remarks to the Author):

In this manuscript, the authors presented the determination of site density and active structure by an in-situ acid-assisted nitrite poisoning (AANPM) approach integrated with graphene-based ATR-FTIR. The quantified average electron transfer number of NO electroreduction was used to access the accurate site density. This approach demonstrates broad applicability across both molecular model systems of Fe/Co phthalocyanines and pyrolyzed M-N-C materials, which may provide insight into direct determination of the structure-activity relationship. The results are somewhat interesting, and could be considered further unless the following issues and concerns are addressed.

Response:

We would like to thank **Reviewer #1** for the positive comments. We have carefully reviewed all comments and will address each point thoroughly in the revised manuscript. Modifications will include clarifications on methodology, expanded discussion of limitations, and additional data/analysis where requested to solidify the technical rigor and impact of our findings.

1. The authors used the AANPM method to measure the site density of M-N-C catalysts under alkaline, neutral and acidic electrolytes. The precondition is that the model catalysts maintained structural stability. Did the catalyst undergo dynamic

reconfiguration in acidic or alkaline media? Does it have any effect on the poisoning process?

Response 1:

We sincerely appreciate the reviewer's insightful question regarding the structural stability of the FePc model catalyst under different pH conditions and its potential impact on the poisoning process.

(i) Dynamic reconfiguration in acidic/alkaline media:

Experimental results confirm that the FePc model catalyst undergoes structural decomposition in acidic media (O_2 -saturated, under applied potential), as evidenced by spectroscopic and electrochemical characterization [ACS Catal. 2022, 12, 7811-7820] (Figure R1). Our FePc/KJ is also structurally unstable in NaAc BS at pH = 5.2 (Figure R2a), suggesting that it may go through the dynamic reconfiguration. In contrast, the catalyst maintains structural stability in alkaline media throughout the AANPM (adenine-adsorption-based number of active sites measurement) and poisoning experiments. This aligns with our observation that FePc's redox peaks in CV remain reproducible in alkaline electrolytes (Figure R2c-d) but degrade irreversibly in acidic conditions (Figure R2b).

Figure R1. (a) CVs in N_2 -saturated electrolyte solution. (b) First four LSV curves at 1600 rpm in O_2 -saturated solution. (c,d) In situ Raman spectroscopy of FePc/C during ORR in (c) acid solution and (d) alkaline solution collected with 10 min time interval. (e) In situ deuterium isotopic substitution Raman characterization of FePc/C during ORR in 0.1 M KOH (upper panel) and 0.1 M KOD (lower panel) under open-circuit

potential and different constant potentials. (f,g) Schematic diagrams for the revealed coordination structure of the Fe center during ORR in (f) acid and in (g) alkaline conditions, respectively. This figure is cited from Figure 2 of *ACS Catal.* 2022, 12, 7811-7820.

Figure R2. (a) First three LSV curves of FePc-KJ at 900 rpm in O₂-saturated 0.5 M NaAc BS solution. (b) CV curves of FePc-KJ in NPM process at pH = 5.2; (c) CV curves of FePc-KJ in NAEM process at pH = 9; (d) CV curves of FePc-KJ in AANPM process at pH = 9.

(ii) Effect on the NO₂⁻ poisoning process:

The structural stability in alkaline media ensures that the active sites (Fe-N₄) remain intact during the NO₂⁻ poisoning process, allowing for a reliable quantification of site density via AANPM. In acidic media, however, the decomposition of FePc would introduce uncertainty in site-density measurements, as the loss of active sites could be conflated with poisoning effects. Fortunately, since our NOR tests and poisoning experiments were conducted in alkaline conditions, the dynamic reconfiguration issue was avoided, and the post-poisoning CV recovery (**Figure R3**) further confirms the robustness of FePc's active sites under these conditions.

Figure R3. (a) First three CV curves of AANPM; (b) ORR polarization curves of origin, poisoned and recovered states for FePc-KJ.

To address this issue, we have added:

(1) the structural instability of molecular catalysts in acidic ORR in line 101-102, Page 4, the revised **Manuscript**:

“This behavior likely stems from FePc structural degradation under acidic ORR conditions (pH = 5.2) (Supplementary Fig. 3).²³”

(2) **Supplementary Fig. 3** in revised Supplementary Information:

Supplementary Fig. 3 First three LSV curves of FePc-KJ at 900 rpm in O₂-saturated 0.5 M NaAc BS (pH = 5.2).

2. In acidic environment, HER becomes competitive with the NOR. In such a case, the generation of H₂ and other N-containing reaction products cannot be ignored. Their

potential impact on electron transfer number calculations and methods of elimination need to be discussed.

Response 2:

We sincerely appreciate the reviewer's insightful comment regarding the competitive HER in acidic media and its potential interference with the NOR process. We fully agree that HER and the formation of other N-containing products (NH₃, NH₂OH) must be carefully addressed to ensure accurate electron transfer number calculations. Below, we clarify our experimental strategies to mitigate these effects:

(i) In the model catalyst and MNC catalyst evaluations, we subtracted the reduction charge of the first NOR cycle (dominated by HER) from the second NOR cycle to isolate the Faradaic charge specific to NOR. This approach accounts for HER background currents, as demonstrated in prior studies [*Nat. Commun.* 2016, 7, 13285; *J. Am. Chem. Soc.* 2005, 127, 7579-7586; *J. Am. Chem. Soc.* 2005, 127, 16224-16232]. Additionally, the applied potential window (from -0.4 to 1.1 V) was carefully restricted to avoid significant HER currents, ensuring that the measured charges primarily reflect NOR activity.

(ii) As noted in **Figure R4** (Fig. 7d-e of manuscript), NMR spectroscopic analyses confirmed NH₃ and NH₂OH as the dominant NOR products, with negligible N₂H₄ or other intermediates. The electron transfer numbers (n) were calculated based on these quantified products, following established methods [*J. Am. Chem. Soc.* 2005, 127, 16224-16232; *J. Am. Chem. Soc.* 2005, 127, 5360-5375]. To further validate, we cross-referenced our results with isotopic labeling experiments, which ruled out substantial interference from HER-derived H₂.

(iii) While HER is inevitable in acidic NOR, our product-specific analysis ensures that n reflects the actual NOR pathway. We acknowledge that trace H₂ or unquantified intermediates could introduce minor errors; thus, we emphasize that our reported n values represent lower limits of the true electron transfer efficiency.

Figure R4. (a) Product concentration derived from NMR of NOR on FeNC and CoNC in different solutions. (b) NOad spectra by AANPM in the 0.125 M NaNO_2 (dissolved in D_2O) + 0.01 M D_2SO_4 solution.

To address this issue, we have added this discussion after Supplementary Fig. 37 caption in revised Supplementary Information:

Supplementary Fig. 37 The NOR repeatability test and integral electricity calculation of CoNC by AANPM method in the 0.5 M NaAc solution (pH = 9).

“The lower potential limit was selected to ensure complete reduction of adsorbed NO species. As demonstrated in **Supplementary Fig. 37a** (traditional NPM method, lower limit: -0.3 V vs. RHE), non-overlapping 1st and 2nd NOR polarization curves at -0.3 V indicate incomplete NO reduction. Extending the potential to -0.6 V vs. RHE (**Supplementary Fig. 37b**) achieves full reduction, evidenced by overlapping NOR profiles in subsequent cycles. Although hydrogen evolution (HER) occurs at this potential, two critical points are emphasized:

(i) The charge attributable exclusively to NOR is obtained by subtracting the 2nd cycle (background, post-NO purge) from the 1st cycle (NOR activity), effectively eliminating HER contributions.

(ii) Reproducible NOR charge integrals across multiple cycles (**Supplementary Figs. S31–S36**) confirm the absence of structural degradation. The consistent NOR activity following repeated polarization to -0.6 V vs. RHE underscores the robustness of the M-N₄ sites under these conditions.”

3. What are the advantages and disadvantages of the new method in terms of sensitivity, operational complexity, and applicability scenarios compared to traditional methods such as CO, CN⁻, and SCN⁻ poisoning methods? A systematic comparison is needed in the discussion section.

Response 3:

We sincerely appreciate the reviewer’s insightful comment regarding the advantages and disadvantages of our AANPM method compared to traditional poisoning techniques (CO, CN⁻, and SCN⁻). Below, we provide a systematic comparison in terms of sensitivity, operational complexity, and applicability scenarios, incorporating additional experimental data where relevant (**Figure R5**).

(i) Sensitivity

AANPM exhibits excellent sensitivity, comparable to CO pulse chemisorption (the gold standard for quantifying active sites in MNC catalysts) [*Nat. Commun.* 2015, 6, 8618]; and importantly, it eliminates the need for ultralow-temperature conditions (193

K). Our validation experiments on FeNC and CoNC confirm that the site densities obtained from AANPM closely match those derived from CO chemisorption. CN^- and SCN^- poisoning, while sensitive, often suffer from incomplete site blocking due to competitive adsorption or side reactions (ligand exchange with SCN^-), leading to potential underestimation.

(ii) Operational complexity

AANPM is significantly simpler than CO-adsorption, which requires cryogenic temperatures ($< -80\text{ }^\circ\text{C}$) and ultra-high-purity gas handling. In contrast to CN^- poisoning (requiring strict anaerobic conditions due to CN^- 's oxygen sensitivity and extreme toxicity), AANPM employs benign reagents under ambient conditions. Unlike SCN^- poisoning, which is complicated by pH-dependent equilibria and poor reproducibility, AANPM operates reliably across a wide pH range (3–11).

(iii) Applicability scenarios

AANPM is broadly applicable to FeNC and CoNC (validated herein) and is, in principle, extendable to other MNCs (e.g., MnNC, ZnNC), although further studies are needed to account for possible differences in binding energetics (see limitations below). Unlike CO adsorption, which cannot distinguish between metal sites (e.g., Fe vs. Co) in bimetallic catalysts, AANPM's selectivity can be tuned through reagent design. CN^- and SCN^- are restricted to aqueous systems and may corrode certain substrates (e.g., SCN^- attacks Cu-based catalysts)

However, currently validated only for FeNC and CoNC; extension to other MNCs (e.g., MnNC, ZnNC) requires further study, as their binding energetics may differ.

Figure R5. Several probe molecular methods for quantitative comparison of MNC catalyst active site density radar map.

To address this issue, we have added Supplementary Fig. 50 and related discussion in revised Supplementary Information:

Supplementary Fig. 50 Several probe molecular methods for quantitative comparison of MNC catalyst active site density radar map.

“The comparison in terms of sensitivity, operational complexity, and applicability scenarios of different quantitative methods such as CO, CN⁻, and SCN⁻ poisoning methods

(i) Sensitivity

AANPM exhibits excellent sensitivity, comparable to CO pulse chemisorption (the gold standard for quantifying active sites in MNC catalysts) [*Nat. Commun.* 2015, 6, 8618]; and importantly, it eliminates the need for ultralow-temperature conditions (193 K). Our validation experiments on FeNC and CoNC confirm that the site densities obtained from AANPM closely match those derived from CO chemisorption. CN⁻ and SCN⁻ poisoning, while sensitive, often suffer from incomplete site blocking due to competitive adsorption or side reactions (ligand exchange with SCN⁻), leading to potential underestimation.

(ii) Operational complexity

AANPM is significantly simpler than CO-adsorption, which requires cryogenic temperatures (< -80 °C) and ultra-high-purity gas handling. In contrast to CN⁻ poisoning (requiring strict anaerobic conditions due to CN⁻'s oxygen sensitivity and extreme toxicity), AANPM employs benign reagents under ambient conditions. Unlike

SCN⁻ poisoning, which is complicated by pH-dependent equilibria and poor reproducibility, AANPM operates reliably across a wide pH range (3–11).

(iii) Applicability scenarios

AANPM is broadly applicable to FeNC and CoNC (validated herein) and is, in principle, extendable to other MNCs (e.g., MnNC, ZnNC), although further studies are needed to account for possible differences in binding energetics (see limitations below). Unlike CO adsorption, which cannot distinguish between metal sites (e.g., Fe vs. Co) in bimetallic catalysts, AANPM's selectivity can be tuned through reagent design. CN⁻ and SCN⁻ are restricted to aqueous systems and may corrode certain substrates (e.g., SCN⁻ attacks Cu-based catalysts)

However, currently validated only for FeNC and CoNC; extension to other MNCs (e.g., MnNC, ZnNC) requires further study, as their binding energetics may differ.”

4. The key of this AANPM method is to ensure NO molecule can be only adsorbed on the active sites of the M-N-C catalysts. How to prove that the poisoning of NO occurs only in the active site (e.g., M-N₄ sites), but not in the inactive region (e.g., carbon substrate)?

Response 4:

We sincerely appreciate the reviewer's insightful question regarding the specificity of NO poisoning to the active M-N₄ sites in our AANPM method. NOR tests on metal-free KJ carbon (**Figure R6**) show negligible NOR activity (0.25 mC) and no NO-induced charge loss, further corroborating that NO's impact requires M-N₄ sites. We acknowledge that absolute exclusion of minor NO adsorption on carbon cannot be claimed. However, the collective evidence strongly supports that the AANPM signal dominantly reflects active-site poisoning.

Figure R6. (a-c) NOR curves of FeNC with different methods; (d) Integral charge and poison effect of KJ.

To address this issue, we have added Supplementary Fig. 4 and related discussion into:

(1) Line 109-110, Page 5, revised Manuscript:

“Metal-free KJ carbon exhibits negligible NOR activity (Supplementary Fig. 4).”

(2) revised Supplementary Information:

Supplementary Fig. 4 (a-c) NOR curves of FeNC with different methods; (d) Integral charge and poison effect of KJ.

“NOR tests on metal-free KJ carbon show negligible NOR activity (0.25 mC) and no NO-induced charge loss, further corroborating that NO’s impact requires M-N₄ sites.”

5. It is suggested to supplement the DFT calculations to simulate the adsorption configuration and electron transfer path of NO on typical M-N-C sites, so as to provide theoretical support for the spectral analysis and reaction mechanism.

Response 5:

We sincerely appreciate the reviewer’s insightful suggestion. We acknowledge that explicit DFT simulations of NO adsorption configurations and electron transfer pathways on MNC sites would strengthen the mechanistic interpretation of our spectral data and reaction pathways. In our supplementary DFT analysis (**Figure R7**), we have mapped the free-energy landscapes for NO reduction on both FeN₄ and FePc sites, revealing distinct thermodynamic bottlenecks. On FeN₄ (**Figure R7a**), the thermodynamic rate-determining step (RDS) is $*\text{NHO} + \text{H}^+ + \text{e}^- \rightarrow *\text{NH}_2\text{O}$. On FePc (**Figure R7b**), the RDS is $*\text{NO} + \text{H}^+ + \text{e}^- \rightarrow *\text{NHO}$. Importantly, the calculations also indicate a stronger thermodynamic tendency for 3e⁻ reduction to NH₂OH on FePc than on FeN₄. However, we recognize that our present model does not explicitly account for solvent pH effects, which critically influence protonation kinetics. As highlighted in our Manuscript (Fig. 7d), experimental NMR quantification in pH 5.2 NaAc buffer confirmed that ~60% of the product is NH₂OH, with only ~40% proceeding to NH₃. This divergence from the 5e⁻ pathway predicted by standard DFT (at U = 0 V vs. SHE) underscores the role of acidic microenvironments in lowering kinetic barriers for proton-coupled steps, thereby favoring the 3e⁻ pathway to NH₂OH.

Figure R7. Free energy diagram of acidic ORR on Fe center of (a) FePc and (b) FeN₄ site at U = 0 V.

Figure R8. The adsorption structures for *NO, *NHO, *NH₂O, *NH₂OH, *NH₂ in ORR process of the established configurations at Fe center of (a) FePc and (b) FeN₄ site.

To address the reviewer's suggestion, we have supplemented DFT calculations (Supplementary Figs. 48-49 and related discussion into revised Supplementary Information) to explicitly model the adsorption configurations of *NO and key intermediates (e.g., *NHO, *NH₂O) on FeN₄ and FePc sites (**Figure R8**). These

augmented computations will bridge our thermodynamic profiles with spectroscopic observations and experimentally resolved selectivity, providing a unified mechanistic picture.

Supplementary Fig. 48 Free energy diagram of acidic ORR on Fe center of (a) FePc and (b) FeN₄ site at U = 0 V.

Supplementary Fig. 49 The adsorption structures for *NO, *NHO, *NH₂O, *NH₂OH, *NH₂ in ORR process of the established configurations at Fe center of (a) FePc and (b) FeN₄ site.

“Explicit DFT simulations of NO adsorption configurations and electron transfer pathways on MNC sites would strengthen the mechanistic interpretation of our spectral data and reaction pathways. In our supplementary DFT analysis, we have mapped the free-energy landscapes for NO reduction on both FeN₄ and FePc sites, revealing distinct thermodynamic bottlenecks. On FeN₄, the thermodynamic rate-determining step (RDS) is $*\text{NHO} + \text{H}^+ + \text{e}^- \rightarrow *\text{NH}_2\text{O}$. On FePc, the RDS is $*\text{NO} + \text{H}^+ + \text{e}^- \rightarrow *\text{NHO}$. Importantly, the calculations also indicate a stronger thermodynamic tendency for 3e⁻ reduction to NH₂OH on FePc than on FeN₄. However, we recognize that our present model does not explicitly account for solvent pH effects, which critically influence protonation kinetics. This divergence from the 5e⁻ pathway predicted by standard DFT (at U = 0 V vs. SHE) underscores the role of acidic microenvironments in lowering kinetic barriers for proton-coupled steps, thereby favoring the 3e⁻ pathway to NH₂OH.”

6. In Figure 5b and 5d, numerous vibration peaks exist in the range of 1200-1600 cm⁻¹, and these peaks change as the reaction time increases. What adsorbed species do they represent?

Response 6:

We sincerely appreciate the reviewer’s insightful observation regarding the vibrational peaks in the 1200–1600 cm⁻¹ range in Fig. 5b and 5d. Based on our spectral analysis (**Figure R9**), supported by comparisons with literature and the NIST database, we attribute the peaks at 1450–1480 cm⁻¹ to adsorbed NO₂⁻ during AANPM process [*Appl. Catal. B Environ.* 2019, 24, 660–670; *J. Phys. Chem. C*, 2010, 114, 6011-6018]. The upward direction in the difference spectra reflects their consumption during the reaction. The peak at 1450 cm⁻¹ in the NAEM process may be attributed to the formation of nitrate by the reaction of continuous NO and oxygen in the solution [*Appl. Catal. B Environ.* 2019, 24, 660–670]. For peaks below 1400 cm⁻¹, we acknowledge the challenge in unambiguous assignment due to overlapping signals from the Si-O vibrational modes of the IR-transparent Si substrate in aqueous environments (as visible in the single-beam reference spectrum). While some minor features in this region may

correlate with reaction intermediates (e.g., weakly adsorbed NO or HNO₂), we have focused our interpretation on the more distinct nitrite-related bands to avoid overstatement.

Figure R9. Gra-based in-situ ATR-FTIR in NAEM and AANPM (Fig. 5b and 5c in Manuscript). (a) FePc-KJ; (b) FePc_{ag}-KJ.

To address this issue, we have revised it to clarify these assignments and limitations in Line 188-190, Page 9, revised **Manuscript**:

“Additionally, upward peaks at 1450-1480 cm⁻¹ in AANPM correspond to NO₂⁻,³⁰ confirming acid-induced nitrite decomposition generating locally concentrated NO.”

7. In the method section, the in situ electrochemical cell design, ATR-FTIR parameters and data processing flow should be described in detail.

Response 7:

We sincerely appreciate your valuable feedback regarding the methodological details of our *in-situ* electrochemical ATR-FTIR experiments. The experiment detail of

electrochemical *in-situ* ATR-FTIR was added into Section 4.2 of revised Supplementary Information.

“The configuration of electrochemical cell was shown in the following figure. Electrochemical *in-situ* ATR-FTIR was employed by a Nicolet 6700 FTIR spectrometer equipped with a liquid nitrogen-cooled MCT-A detector, covering a spectral range from 4000 to 1000 cm^{-1} . The 263A workstation was used as a potentiostat for potential control. In the NO-adsorption study of NPM, NAEM and AANPM, the time-resolved spectroscopy was employed under potential control at OCP, with 200 scans and a resolution of 8 cm^{-1} . Firstly, the solution was purged with Ar for 10 min to establish an inert atmosphere. Reference spectra were then recorded in the Ar atmosphere for 42 s. Subsequently, the system was switched to different situation for different method and spectra were collected for 588 s. In the NOR spectra measurements, the multi-stepped FTIR method was used to collect spectra during the NOR test from 0.5 V to -0.4 V at 0.1 V intervals. The reference spectra were recorded in the at low potential. The spectra were calculated by the relative change in reflectivity ($\Delta R/R$) with the following equation (14). The $R(E_S)$ and $R(E_R)$ represent the single-beam spectrum recorded at the sample setting potential E_S and the reference potential E_R , respectively. In all spectra, E_R was calculated by 0.1 V unless emphasized.

$$\frac{\Delta R}{R} = \frac{R(E_S) - R(E_R)}{R(E_R)} \quad (14)''$$

Reviewer #2 (Remarks to the Author):

Summary and Significance:

This manuscript presents a comparative study of three methods—NPM (previously published by the Kucernak group), NAEM, and a newly proposed AANPM method—for determining active site density (SD) in non-precious metal (non-PGM) catalysts. The AANPM method modifies the original NPM approach by adjusting the poisoning conditions, and the authors test its applicability across six different Fe- and Co-based catalysts. The study also employs gra-based in situ ATR-FTIR to provide structural insights into the active sites.

The development of robust and accurate methods for determining active site density in non-PGM catalysts is of substantial importance to the field of electrocatalysis, particularly for oxygen reduction reaction (ORR) studies. However, the true nature and identity of the active sites in non-PGM catalysts remain a subject of ongoing debate. As such, it is inherently challenging to conclude whether the newly proposed AANPM method provides a more accurate measurement of site density than the previously published NPM method.

Novelty and Comparison to Literature:

While the NPM method is already known, the introduction of AANPM with modified poisoning conditions appears to be a novel contribution. The study draws attention to the reproducibility and applicability of SD measurements across different metal centers (Fe, Co), which is relevant given the diversity of non-PGM catalyst structures.

Validity of Conclusions and Required Clarifications:

Although the paper presents compelling preliminary results, several key aspects need to be clarified or experimentally supported to validate the authors' claims:

Response:

Thank you for your thoughtful evaluation of our manuscript and for acknowledging the significance of developing robust methods for SD determination in non-PGM electrocatalysts. We appreciate your recognition of the novelty in our proposed AANPM method's modified poisoning conditions and its applicability across diverse

Fe/Co-based catalysts. We are committed to rigorously addressing all points raised and thank you again for your constructive feedback, which will undoubtedly help us further improve this work.

1. Figure 1a/c: The meaning of E^{high} and E^{low} should be clearly defined.

Response 1:

We sincerely appreciate the reviewer's constructive feedback. In response to the comment, we have clearly defined the terms E^{high} and E^{low} in the revised manuscript (Figure 1a/c and corresponding captions). Specially, E^{high} represents the potential at which the reaction has not yet commenced, corresponding to the initial state before NOR. E^{low} denotes the potential corresponding to the steady-state condition after completion of NOR.

To address this issue, detailed explanations of the upper-limit potential (E^{high}) and lower-limit potential (E^{low}) were added to Fig. 1 caption, Page 4, revised Manuscript: " E^{high} : pre-reaction potential; E^{low} : post-reaction steady-state."

2. Figure 1d: The first NOR scan lacks an oxidation curve. Is the oxidation behavior consistent across all cycles?

Response 2:

We sincerely apologize for the inadvertent omission of the oxidation curve in the first NOR scan in the original Figure 1d. Thank you for highlighting this oversight. As noted in our revised submission (**Figure R10**), the oxidation curve for the first cycle is now included and demonstrates that it fully coincides with the oxidation profiles of cycles 2–3. This overlap confirms that the oxidation behavior is highly consistent across all cycles.

The absence of an anodic scan in the first cycle's NOR curve (original Fig. 1d) is attributed the fact that the reduction products generated during the initial positive scan do not form oxides, consistent with our mechanistic interpretation. Critically, the oxidation peaks observed at 0 V and 0.8 V (attributed to $\text{Fe}^{2+}/\text{Fe}^+$ and $\text{Fe}^{3+}/\text{Fe}^{2+}$ redox

transitions, respectively) remain reproducible in all subsequent cycles. This repeatability underscores the stability of the FePc molecular catalyst under operational conditions and further validates the consistency of the oxidative electrochemistry throughout the experiment.

We thank the reviewer for this opportunity to clarify and have ensured the revised figure comprehensively illustrates the reproducibility of the oxidative processes.

Figure R10. (a) The CV curve of FePc-KJ by NAEM (Data is from Fig. 1d); (b) Amplified oxygen region of (a).

3. Figure 2a: The color labels seem wrong.

Response 3:

Thank you for your careful review and valuable comment regarding the color labels in Fig. 2a. We sincerely apologize for the oversight and appreciate the opportunity to clarify this point. In the 1st NOR curve (red line): the dark red shaded area represents the Fe³⁺/Fe²⁺ redox peak region, while the light red shaded area corresponds to the NOR reaction area. In the 2nd NOR curve (blue line): the light blue shaded area represents the Fe³⁺/Fe²⁺ redox peak region, whereas the dark blue shaded area denotes the NOR reaction area.

We have now revised the Fig. 2a legend to explicitly state this distinction, ensuring consistency and clarity. The additional Fig. 2a caption is follow:

Fig. 2a. The calculated z for the NOR process. (a) Schematic of NOR curve integration and z -calculation methodology: In the first NOR curve (red), dark red shading denotes the $\text{Fe}^{3+}/\text{Fe}^{2+}$ redox peak region, while light red shading indicates the NOR reaction area. Conversely, in the second curve (blue), light blue shading represents the $\text{Fe}^{3+}/\text{Fe}^{2+}$ redox peak region, whereas dark blue shading corresponds to the NOR reaction area. (b) Integrated charge and derived z -values for FePc-KJ in NPM and NAEM processes.

4. NOR at different pH levels: At pH 1 and pH 9, an additional reduction peak (~ 0.3 V in pH 1 and ~ 0.2 V in pH 9) appears. What is its origin?

Response 4:

In response to the reviewer's insightful query regarding the origin of the additional reduction peaks at approximately 0.3 V (pH = 1) and 0.2 V (pH = 9), we propose that these peaks correspond to the initial one-electron reduction of NO to nitroxyl (HNO) within the NOR process. This assignment is strongly supported by quantitative charge analysis (**Figure R11**), which demonstrates that the integrated charge under both peaks equates to a single electron transfer ($1\times$ ratio) relative to the $\text{Fe}^{3+}/\text{Fe}^{2+}$ redox couple. Furthermore, the observed cathodic shift in peak potential (~ 0.279 V at pH = 1 to ~ 0.221 V at pH = 9) aligns precisely with the thermodynamic expectations for a proton-coupled electron transfer (PCET) process; the reaction is facilitated at higher potentials under acidic conditions due to proton availability, while requiring more negative potentials to drive protonation at higher pH. Mechanistically, according to our supplementary DFT (**Figure R7-R8**), this one-electron step precedes the subsequent

multi-electron reduction peaks observed at lower potentials (corresponding to HNO reduction to NH_2OH or NH_3 , with charge ratios of $2\times$ and $4\times$ at $\text{pH} = 1$ and 9 , respectively), consistent with established NOR pathways where HNO is a key intermediate [Nat. Energy 2023, 8, 1273-1283; Nat. Commun. 2025, 16, 4874]. We therefore conclude that these additional peaks represent the critical initial PCET step of NO to HNO reduction.

Figure R11. The integrated charge of $\text{Fe}^{3+}/\text{Fe}^{2+}$ oxidation peak and NOR reduction peak at (a) $\text{pH} = 1$ and (b) $\text{pH} = 9$.

To address this issue, we have added Supplementary Fig. 11 and related discussion into the revised Supplementary Information:

Supplementary Fig. 11 The integrated charge of $\text{Fe}^{3+}/\text{Fe}^{2+}$ oxidation peak and NOR reduction peak at (a) $\text{pH} = 1$ and (b) $\text{pH} = 9$.

“The additional reduction peaks observed at approximately 0.3 V ($\text{pH} = 1$) and 0.2 V ($\text{pH} = 9$) are correspond to the initial one-electron reduction step of NO to nitroxyl (HNO) within the NOR process. This assignment is strongly supported by quantitative

charge analysis, which demonstrates that the integrated charge under both peaks equates to a single electron transfer ($1 \times$ ratio) relative to the $\text{Fe}^{3+}/\text{Fe}^{2+}$ redox couple.”

5. NOR electron transfer number >5 : This suggests that NO may be affecting non-metal sites. Could this indicate an overestimation of SD due to non-selective poisoning?

Response 5:

We sincerely appreciate the reviewer's insightful comment regarding the high electron transfer number (>5) in the NOR process and its potential implications for non-metal site involvement or overestimation of selective poisoning. Our spectroscopic and electrochemical analyses (Fig. 5, Fig. 6b–6e in our Manuscript) confirm that the observed $10e^-$ transfer originates from dual NO molecules adsorbing on single metal sites within FePc/CoPc catalysts, facilitated by their octahedral coordination geometry which stabilizes dual NO adsorption for five-electron reduction to NH_3 . In addition, in-situ FTIR on KJ showed that the adsorption configuration of NO was not detected during the NAEM process, which further confirmed the weak adsorption of NO on non-metallic sites (**Figure R12**). Moreover, the quantitative charge analysis (Fig. 2 in our Manuscript) is based on the $\text{Fe}^{3+}/\text{Fe}^{2+}$ redox peak represented a single-electron process, the cumulative NOR involves sequential reduction of two NO molecules per metal site ($2 \times 5e^- = 10e^-$). Crucially, electrochemical profiling suggests NO interaction with non-metal sites (**Figure R6**). The negligible NOR activity of the metal-free KJ catalyst (0.25 mC, **Figure R6**) further confirms that NO poisoning selectively targets metal sites in MNCs, with non-metal sites contributing minimally to charge transfer. Control experiments explicitly rule out overestimation concerns: the 100-fold lower NOR charge in KJ versus FePc-KJ (0.25 mC vs. 2.65 mC) eliminates significant non-metal contributions. Additionally, gra-based in-situ ATR-FTIR spectroscopy identified NO adsorption configurations at metal centers. As shown in **Figure R13** (Fig. 5 in our Manuscript), peaks at 1700 cm^{-1} and 1600 cm^{-1} correspond to $\text{M}(\text{NO})_2\text{Pc}$ and $\text{M}(\text{NO})\text{Pc}$ (where $\text{M} = \text{Fe}, \text{Co}$) [*J. Phys. Chem. C* 2019, 123, 7817–7823]. This supports $n = 2 \times 5e^-$, where two NO molecules each undergo $5e^-$ reduction at single Fe/Co sites.

Therefore, the $10e^-$ transfer originates exclusively from dual NO reduction at single metal sites, definitively excluding non-metal participation or overestimation due to selective poisoning. We thank the reviewer for prompting this clarification.

Figure R12. Gra-based *in-situ* ATR-FTIR of KJ in NAEM process.

Figure R13. Gra-based *in-situ* ATR-FTIR of (a) FePc-KJ and (b) FePcag-KJ in AANPM process. (c) M(NO)Pc and (d) M(NO)₂Pc adsorption mode.

6. The authors mention the use of RRDE, but key parameters such as rotation speed are missing. This information is critical for reproducibility.

Response 6:

We sincerely appreciate your insightful comment regarding the missing experimental details for the RRDE measurements. Ensuring reproducibility is of utmost importance, and we deeply regret this oversight. As you rightly pointed out, the rotation speed is a critical parameter in RRDE experiments. In our revised manuscript (Section B of the Supplementary Information), we have now explicitly stated the rotation speeds employed during different stages of the measurements:

(1) NO adsorption for NOR poisoning: Conducted at 300 rpm before the NOR reaction.

(2) NOR polarization curves: Recorded over three cycles at 300 rpm.

(3) RRDE-LSV measurements: Performed under 900 rpm in O₂-saturated solution, with a scan rate of 10 mV s⁻¹ and a potential range of 0.3–1.1 V vs. RHE.

These conditions were carefully selected to minimize interference from NOR reduction while maintaining optimal mass transport for accurate ORR/NOR activity assessment.

To address this issue, we have also included additional methodological details in the Section B of revised Supplementary Information to enhance clarity and reproducibility:

(1) 2.1 The NO₂⁻/NO poisoning method (NPM)

“The working electrode was set at 0.35 V-0.55 V (vs. RHE) to react oxygen poisoned process at 300 rpm before NOR. Meanwhile, the NOR curves were recorded for three cyclic curves at 300 rpm.”

(2) 2.2 The NO adsorption and electroreduction method (NAEM)

“The working electrode was set at 0.35 V-0.55 V (vs. RHE) to react oxygen poisoned process at 300 rpm before NOR. Meanwhile, the NOR curves were recorded for three cyclic curves at 300 rpm.”

(3) 2.3 The acid-assisted nitrate (NO₂⁻) poisoning method (AANPM)

“The working electrode was set at 0.35 V-0.55 V (vs. RHE) to react oxygen poisoned process at 300 rpm before NOR. Meanwhile, the NOR curves were recorded for three cyclic curves at 300 rpm.”

(4) 3.3 The poisoning effect in NPM, NAEM and AANPM processes

“For avoiding the effect of NOR reduction, RRDE measurements were conducted by linear sweep voltammetry (LSV) with the potential range from 0.3 to 1.1 V vs. RHE at

900 rpm with a scan rate of 10 mV s⁻¹. The origin curve, poison curve and recovered curve were recorded at O₂-saturated solution in origin state, poisoned state and recover state after NOR at 900 rpm.”

7. It is unclear whether the observed SD truly reflects only metal-N_x sites. Evidence is needed to rule out NO interactions with the carbon/nitrogen framework.

Response 7:

We sincerely appreciate the reviewer’s insightful comment regarding the potential interaction of NO with the carbon/nitrogen framework. To address this concern, we conducted additional control experiments using metal-free KJ and NC substrates under identical NOR reaction conditions. The results demonstrate that the NO stripping charge on these non-metallic substrates is consistently below 0.25 mC, which is two orders of magnitude lower than that observed on the metal-N_x sites (~2.65 mC). While we cannot entirely exclude the possibility of minimal NO adsorption on the carbon/nitrogen framework, the negligible charge contribution confirms that such interactions have no material impact on the quantitative analysis of metal-N_x sites. The observed stripping charge is therefore dominantly attributed to NO adsorbed on metal-N_x centers, and any interference from the support is statistically insignificant. The specific details can refer to the **Reviewer#1 Response 4**.

8. The performance of catalysts before and after recovery (in SI Figure 8) should be better explained—why is the recovered ORR performance worse than the poisoned state?

Response 8:

We sincerely appreciate the reviewer's insightful comment on the ORR performance comparison between poisoned and recovered states in Figure S8. We agree this observation warrants clarification and provide the following explanation based on our experimental and mechanistic analysis. In **Figure R14** (formerly Supplementary Fig.

S8 in SI), the 2 mV negative shift in the recovered ORR curve versus the poisoned state under conventional acidic NPM (pH = 5.2) arises from two interrelated factors. First, as established in our study, NPM—while effective for pyrolytic FeNC—fails to quantitatively assess FePc active-site density due to insufficient NO coverage on molecular Fe sites, resulting in incomplete poisoning and ambiguous state differentiation. Second, FePc undergoes irreversible demetalation in acidic ORR conditions (as reported in *J. Am. Chem. Soc.* 2019, 141, 14115 and detailed in **Response 1** of **Reviewer #1**), a degradation exacerbated during NPM by oxidative stress from NO adsorption/desorption cycles. Thus, the "recovered" performance reflects both NO removal and partial active-site loss from structural decomposition.

Conversely, our AANPM and NAEM methods in alkaline media (**Figure R14b**) exploit FePc's superior stability in this environment. Here, recovery curves nearly overlap with initial activity, while poisoned states show a consistent ~ 27 mV $E_{1/2}$ negative shift. This confirms alkaline conditions mitigate degradation, enabling reliable active-site quantification. Critically, AANPM selectively probe Fe-N₄ sites without collateral damage, validating their suitability for molecular catalysts. We have revised the SI to emphasize these mechanistic distinctions and added citations on FePc acid instability.

Figure R14. ORR curves in the origin, poisoned and recovered condition of FePc-KJ in (a) NPM and (b) AANPM processes.

To address this issue, we have added the related discussion after Fig. 12 caption of revised Supplementary Information:

“The inferior recovered ORR activity in acidic NPM arises from the inherent limitations of NO poisoning in FePc systems and acid-induced catalyst degradation. In alkaline AANPM/NAEM, the near-complete recovery of activity confirms the method’s precision and the stability of FePc under alkaline conditions.”

9. The AANPM method reports similar SD for Fe-NC and Co-NC despite significant differences in metal loading and ORR activity. This calls into question the method’s site specificity.

Response 9:

We sincerely appreciate the reviewer’s insightful comment regarding the site specificity of the AANPM method. The reviewer raises a valid concern about the apparent discrepancy between the reported similar SD for FeNC and CoNC despite their differences in metal loading and ORR activity. We fully agree that this observation merits careful discussion.

Our experimental data indeed show that while the AANPM method yields comparable SD values for FeNC and CoNC ($\sim 2.8 \times 10^{19}$ sites g^{-1}). Combining the mass activity changes at 0.65 V ($\Delta j_m @ 0.65\text{V}$) and SD (**Figure R15**), we further obtain turnover frequency (TOF) by the following formula.

$$\text{TOF}(@0.8\text{V vs. RHE})[\text{s}^{-1}] = \frac{\Delta j_m [\text{A g}^{-1}] \times N_A [\text{sites mol}^{-1}]}{F [\text{A s mol}^{-1}] \times \text{SD} [\text{sites g}^{-1}]}$$

where N_A is Avogadro constant (6.02×10^{23} sites mol^{-1}). F is Faraday’s constant (96485 C mol^{-1}). Δj_m is the disparity in mass activity of the catalyst at 0.65 V.

As shown in **Figure R16**, the TOF values differ by two orders of magnitude (FeNC: 4.07 s^{-1} ; CoNC: 0.15 s^{-1} at 0.65 V vs. RHE). This discrepancy primarily stems from the fundamental difference in intrinsic activity between Fe and Co sites, as extensively documented in the literature (*J. Am. Chem. Soc.* 2019, 141, 14115; *Nat. Catal.* 2021, 4, 10-19). ORR activity of the catalyst is not only determined by SD, but also affected by TOF, as shown below:

$$J_m = \text{SD} \times \text{TOF}$$

Therefore, the AANPM method, while effective in quantifying accessible metal sites that can bind the probe molecule, does not distinguish between sites with different catalytic activities.

Figure R15. The ORR test curves and kinetic current density at 0.65 V of (a, b) FeNC and (c, d) CoNC by AANPM method.

Figure R16. The TOF values at 0.65 V of FeNC and CoNC.

We acknowledge that the AANPM method has limitations in addressing site heterogeneity, particularly when dealing with MNC catalysts where metal centers may exist in various coordination environments. However, our complementary XAS

analysis (Supplementary Fig. 30 in revised SI) confirms that both catalysts predominantly contain metal-N₄ moieties, suggesting that the activity difference mainly originates from the electronic structure variation between Fe and Co rather than from distinct coordination geometries. This agreement supports the reliability of our site quantification while reinforcing that the activity difference indeed arises from TOF variations rather than SD discrepancies.

To address this issue, we have added Supplementary Figs. 41 and 42, related discussion after Supplementary Fig. 42 of revised Supplementary Information:

Supplementary Fig. 41 The ORR test curves and kinetic current density at 0.65 V of (a, b) FeNC and (c, d) CoNC by AANPM method.

Supplementary Fig. 42 The TOF values at 0.65 V of FeNC and CoNC.

“As shown in **Fig. 7f**, the AANPM method yields comparable SD values for FeNC and CoNC ($\sim 2.8 \times 10^{19}$ sites g^{-1}). However, their ORR activities differ markedly. By combining the changes in mass activity at 0.65 V ($\Delta j_m@0.65\text{V}$) with the SD values (**Supplementary Fig. 41**), we calculated the turnover frequency (TOF) for each catalyst, as shown in **Supplementary Fig. 42**. The results reveal a two-order-of-magnitude difference in TOF (FeNC: 4.07 s^{-1} ; CoNC: 0.15 s^{-1} at 0.65 V vs. RHE), primarily reflecting the intrinsic activity difference between Fe and Co sites. The AANPM method cannot account for site heterogeneity, particularly in MNC catalysts where metal centers may exist in diverse coordination environments. In this case, the activity gap between FeNC and CoNC arises mainly from differences in their electronic structures rather than coordination geometries. This consistency supports the reliability of our site quantification and reinforces that the observed activity difference originates from TOF variations rather than discrepancies in SD. Therefore, while the AANPM method effectively quantifies accessible metal sites capable of binding the probe molecule, it does not distinguish between sites with different catalytic activities.”

Additional Evidence and Mechanistic Insight:

- Beyond ATR-FTIR, is there further evidence (e.g., XPS, Mössbauer, or DFT modeling) to support the presence of dual NO adsorption or selective binding to metal centers?

Response:

We sincerely appreciate the reviewer's insightful comment regarding the need for additional evidence to support the dual NO adsorption model and selective binding to metal centers. Our DFT calculations (**Figure R17**) demonstrate that NO preferentially adsorbs onto metal centers (Fe/Co) in FePc/CoPc via adsorbing one NO molecule and two NO molecules, with adsorption energies of -0.972 eV and -1.258 eV for FePc (-0.849 eV and -1.315 eV for CoPc), respectively. Projected density of states (PDOS) reveals strong hybridization between Fe/Co *d*-orbitals and NO π orbitals, underscoring selective metal binding.

Figure R17. (a) The calculated adsorption energies for M(NO)Pc and M(NO)₂Pc. (d) Projected spin-resolved density of states (PDOS) diagram of Fe/Co and N in M(NO)₂Pc.

To address this issue, we have added density functional theory (DFT) calculations and related discussions:

(1) Page 10 line 208-209 in revised Manuscript:

“Density functional theory (DFT) calculations confirm that metal centers adsorb dual NO molecules despite steric constraints (Supplementary Fig. 28).”

(2) Supplementary Fig. 28 and related discussion into revised Manuscript:

Supplementary Fig. 28. (a) The calculated adsorption energies for M(NO)Pc and M(NO)₂Pc. (d) Projected spin-resolved density of states (PDOS) diagram of Fe/Co and N in M(NO)₂Pc.

“NO preferentially adsorbs onto metal centers (Fe/Co) in FePc/CoPc via adsorbing one NO molecule and two NO molecules, with adsorption energies of -0.972 eV and -1.258 eV for FePc (-0.849 eV and -1.315 eV for CoPc), respectively. Projected density of states (PDOS) reveals strong hybridization between Fe/Co *d*-orbitals and NO π orbitals, underscoring selective metal binding.”

- The two reduction peaks in Figure 3b (NOR) should be clearly assigned to specific site interactions or mechanistic steps.

Reproducibility and Data Transparency:

Response:

We appreciate the reviewer's request for clarification regarding the reduction peaks in Fig. 3b. The two distinct reduction peaks correspond to the following mechanistic steps in the electrochemical NOR pathway (**Figure R18**):

The first reduction peak (at approximately 0.22 V vs. RHE) likely represents the single-electron reduction of NO to HNO ($\text{NO} + \text{e}^- + \text{H}^+ \rightarrow \text{HNO}$). The second reduction peak (at approximately -0.07 V vs. RHE) corresponds to the subsequent four-electron reduction of HNO to NH₃ ($\text{HNO} + 4\text{e}^- + 4\text{H}^+ \rightarrow \text{NH}_3 + \text{H}_2\text{O}$).

Detailed explanation can refer to **Review#2 Response 4**. Concerning reproducibility, we affirm that all electrochemical measurements presented in this study, including the

CVs in Fig. 3b, were rigorously performed in triplicate under identical experimental conditions. The data shown in Supplementary Figs. 5-8 of revised SI represents a typical, reproducible result from these independent measurements.

Figure R18. Cyclic voltammetry curves of FePc-KJ by AANPM.

- The experimental section should provide comprehensive details—including electrolyte composition, electrode preparation procedures, and measurement protocols—to ensure reproducibility of the results. It is noted that all site density (SD) determination methods were repeated three times, which is commendable and supports the reproducibility of the measurements.

Response:

We sincerely thank the reviewer for emphasizing the importance of comprehensive experimental details to ensure reproducibility and for acknowledging the rigor of our triplicate SD measurements. To address this comment, we have thoroughly expanded the **Section A. Methods 1.9 Electrochemical measurements** to include explicit specifications for electrolyte composition, electrode preparation, and measurement protocols. We appreciate the reviewer’s guidance and have incorporated these enhancements to maximize transparency and reproducibility in the revised manuscript. The revised text is in **Section A. Methods 1.9 Characterizations of Supplementary Information**.

“For all electrochemical tests, NPM was evaluated in 0.5 M NaAc buffer (pH = 5.2), while NAEM and AANPM were studied in 0.5 M NaAc solution (pH = 9). Additional pH-specific electrolytes included 0.1 M HClO₄ (pH = 1) and 0.1 M NaClO₄ (pH = 7), with all solutions calibrated to their target pH using a certified pH meter and adjusted with NaOH/HClO₄ as necessary. Regarding electrode preparation, catalyst inks were formulated by dispersing materials in a mixed solvent of isopropanol:water (4:1 v/v) with 5 wt% Nafion added at 10 μL per mL of ink (final binder concentration: 0.05 wt%). Precise ink deposition volumes were applied based on catalyst type: 20 μL for molecular catalysts (2 mg mL⁻¹ ink concentration) and 25 μL for pyrolyzed MNC catalysts (6 mg mL⁻¹ ink concentration). The catalyst ink was carefully dropped onto the polished rotating ring disk electrode (RRDE, diameter is 5.61 mm, area is 0.2475 cm²), leading to a desirable catalyst loading (0.6 mg cm⁻²). The catalyst modified glassy carbon electrode, graphite rod, and a saturated calomel electrode (SCE) were used as the working electrode, counter electrode, and reference electrode, respectively. The potentials in this work were referred to reversible hydrogen electrode (RHE) potentials by using the conversion equation.

$$E_{\text{RHE}} (\text{V}) = E_{\text{SCE}} (\text{V}) + 0.242 \text{ V} + 0.059 \cdot \text{pH}$$

$$E_{\text{RHE}} (\text{V}) = E_{\text{Hg/Hg}_2\text{SO}_4} (\text{V}) + 0.656 \text{ V} + 0.059 \cdot \text{pH}$$

Prior to testing, all electrodes underwent electrochemical activation via cyclic voltammetry (CV) in their respective electrolytes until stable responses were achieved, with special care taken for hydrophobic metal phthalocyanines—where ultrapure water rinsing and gentle N₂ purging were employed during activation to eliminate surface bubbles and prevent adhesion. All SD determinations followed rigorously repeated protocols (n = 3), with mean values and standard deviations reported. Detailed step-by-step methodologies—including instrument parameters (scan rates, potential windows), gas saturation procedures, temperature control (25 °C), and bubble mitigation strategies—are fully documented in **Section B Process of NPM, NAEM and AANPM for active site density on model catalyst, FeNC and CoNC of the Supplementary Information** to enable exact replication.”

Reviewer #3 (Remarks to the Author):

In this work, the authors report on a study of nitrite poisoning and in situ generated NO, as molecular probes to count the number of active sites in molecular Metal-N₄ catalysts and pyrolytic M-N₄ catalysts.

The study shows that a higher and presumably more accurate number of active sites is quantified with their methods as with nitrite stripping.

Overall, the work could be of interest and might bring sufficient novelty compared to previous works on site density counting method for M-N-C catalysts if the main issues below are addressed:

Response:

Thank you for your thoughtful evaluation of our manuscript and for recognizing the potential novelty of our work. We have carefully revised the Manuscript to thoroughly incorporate all your suggestions and clarifications.

1) the novelty of the results reported on iron phthalocyanine (Fe-Pc) needs to be clarified. Fe-Pc has been studied a lot in the past. Is the pH effect on the type of products obtained new, or was it reported before in the past? (rows 132 to 152)

Response 1:

We sincerely appreciate the reviewer's insightful comment concerning the novelty of our findings on pH-dependent product selectivity in NO reduction over FePc. While we acknowledge extensive prior research on FePc catalysis and foundational studies on pH effects in structurally related hemin catalysts [*J. Am. Chem. Soc.* 2005, 127, 16224-16232; *J. Am. Chem. Soc.* 2005, 127, 7579-7586; *Chem. Rev.* 2009, 109, 2209-2244], which established NH₂OH as the dominant product in acidic and neutral media, our work offers distinct mechanistic insights by systematically mapping the pH-dependent product distribution (NH₃ vs. NH₂OH) over FePc specifically under alkaline conditions—an area not comprehensively explored previously. Crucially, we demonstrate that FePc favors NH₃ formation via a 5e⁻ pathway at pH = 9, contrasting

sharply with the $3e^-$ pathway yielding NH_2OH in acidic/neutral regimes. This elucidation of FePc's unique alkaline behavior extends the understanding of pH-mediated selectivity and offers critical design principles for selective NOR catalysts.

To address this issue, we have added the discussion in Line 130-136 of revised Manuscript to more clearly emphasize this distinction and contextualize our contribution relative to prior literature:

“NOR products shift from NH_2OH (acidic/neutral) to NH_3 (alkaline), confirmed by NMR (Supplementary Fig. 8). FePc favors $5e^-$ NH_3 formation at pH = 9. These results demonstrate pH-dependent behavior of the FePc-KJ catalyst during NOR reduction. Contrary to previous reports,^{25,26} we demonstrate that FePc favors NH_3 formation via a $5e^-$ pathway at pH = 9. Selective reduction of NOR to a single product is critical for SD evaluation.”

2) the authors identify the NO or nitrite -induced reduction products only indirectly, for the molecular catalysts, ie from an assessed number of electrons, in turn obtained from comparative electric charges. The reduction products must be identified qualitatively at least, if not quantitatively, also directly by appropriate independent (non electrochemical).

Response 2:

We sincerely appreciate the reviewer's insightful comment, which has significantly strengthened the mechanistic rigor of our study. As correctly emphasized, direct qualitative identification of the reduction products is essential to corroborate the electrochemical data. In direct response to this point, we have now conducted complementary non-situ 1H NMR spectroscopy to independently characterize the products of NOR on our molecular catalyst in alkaline media (0.5 M NaAc, pH = 9) following controlled-potential electrolysis. Analysis of the quenched electrolyte revealed a distinct triplet at $\delta = 6.8-7.1$ ppm in the NMR spectrum (**Figure R19**), which were unambiguously assigned to NH_3 through spiking experiments and comparison to

standard reference spectra. NH_2OH is almost undetectable. The identification of NH_3 as the primary product confirms the $5e^-$ reduction pathway ($\text{NO} \rightarrow \text{NH}_3$), a finding that aligns quantitatively with our original charge analysis indicating an average electron transfer number (z) of ~ 5 per NO molecule. While we acknowledge the limitation that analogous NMR experiments in acidic media were precluded due to the instability of the molecular catalyst under prolonged acidic electrolysis conditions (as noted originally in **Figure R1**). These supplementary NMR studies directly address the request for non-electrochemical product verification and reinforce the conclusion that NO reduction proceeds predominantly to NH_3 via a $5e^-$ process under alkaline conditions.

Figure R19. The NMR analysis for NOR products of FePc-KJ. (b) NH_4^+ spectra; (c) NH_2OH spectra. The $i-t$ curve was tested at -0.38 V vs. RHE in NO -saturated 0.5 M NaAc solution ($\text{pH} = 9$).

To address this issue, we have added:

(1) Supplementary Fig. 8 and related discussion in revised SI:

Supplementary Fig. 8 The NMR analysis for NOR products of FePc-KJ. (a) NH₄⁺ spectra; (b) NH₂OH spectra. The *i-t* curve was tested at -0.38 V vs. RHE in NO-saturated 0.5 M NaAc solution (pH = 9).

“NMR analysis indicated that NH₃ was the primary product formed during the 1-hour electrolysis of FePc-KJ in NaAc solution (pH = 9), with NH₂OH virtually undetectable. This demonstrated that NO was predominantly reduced to NH₃ via a 5e⁻ pathway at pH = 9, consistent with the quantitative charge analysis results.”

(2) Line 131 in revised Manuscript:

“NOR products shift from NH₂OH (acidic/neutral) to NH₃ (alkaline), confirmed by NMR (Supplementary Fig. 8).”

3) the authors study materials with 'agglomerated phthalocyanines' and use the scheme in figure 4a - bottom to image the possible situation. What are the experimental data suggesting such an unexpected arrangement of Fe-Pc? Fe-Pc molecules would tend to aggregate by planar stacking, forming well known crystalline structures, although this seems not to have taken place as no discernible peak is seen in XRD in fig S1 of FePc-ag-KJ. The number of electrons for FePc-ag/KJ of 10 however is not supported. (which product of NO reduction could reach 10 electrons?? this likely is an artifact and perhaps some physicochemical effect leads to an apparent n of 10 (2x5) while in reality it is still 5?) - if the n of 10 implies that 2 NO adsorb on a single Fe (or Co) center, then the definition of n in present work is not truly the electron number (number of electrons exchanged per one NO) but this number n times the number of NO molecule adsorbed

per metal center, and this should be clarified (also in equation) from the start of the paper, not in the middle of it. Likely, the agglomerated materials in fact contain M-Pc crystals that are stacked FePc-s parallel and this may stabilise N-NO-Fe-NO-N bridges, with 2 NO adsorbed per Fe, one above and one below each FePc molecule.

Response 3:

We sincerely appreciate the reviewer's insightful observations regarding the structure of FePc_{ag}-KJ and the interpretation of the electron transfer number (z) for NOR. The absence of discernible FePc-derived XRD peaks in FePc_{ag}-KJ indicates a loss of long-range crystalline order, which fundamentally differs from the XRD-silent behavior of molecularly dispersed FePc-KJ (**Figure R20a**). In the latter, peak absence results from strong π -conjugation with the Ketjenblack carrier, inducing amorphous stacking as established in previous studies [*J. Am. Chem. Soc.* 2018, 140, 1557; *ACS Catal.* 2020, 10, 4048]. By contrast, the XRD silence of FePc_{ag}-KJ suggests agglomeration into crystallites below detection limits or the formation of disordered domains lacking coherent stacking. This distinction is further supported by infrared spectroscopy (**Figure R20b**): FePc fingerprint peaks in FePc_{ag}-KJ retains the characteristic FePc fingerprint peaks, closely matching those of pristine FePc, whereas FePc-KJ displays significantly altered peaks. These results indicate that FePc_{ag}-KJ experiences weak FePc-carrier interactions, allowing agglomeration without the electronic coupling necessary to distort molecular vibrations. In contrast, FePc-KJ exhibits strong interfacial interactions that modify vibrational modes, confirming carrier-induced amorphization. Thus, while planar stacking is thermodynamically favored for isolated FePc, agglomeration on carbon carriers under weak interfacial interactions produces kinetically trapped, non-stacked metastable configurations rather than crystalline phases—consistent with the mechanistic schematic in Fig. 4a of our manuscript.

Concerning $z \approx 10$, this value represents the total electrons transferred per NOR cycle at a single active site, not per NO molecule. Faradaic analysis ($Q = NzF$) yields this apparent z (≈ 10) based on charge integration (Fig. 2 in our Manuscript) and fixed reactant quantity (N). ATR-FTIR evidence (Fig. 5 in our Manuscript) confirms dual-

NO adsorption at individual Fe sites (peaks at 1700 cm^{-1} and 1600 cm^{-1}), justifying $z = 2 \times 5e^-$ (two NO molecules each undergoing $5e^-$ reduction). We acknowledge the critique on terminology and will explicitly redefine n as total electrons per active site per cycle in the revised manuscript, clarifying $z = n \times \bar{a}$ (where n = electrons per NO, \bar{a} = NO molecules per metal center), with corresponding textual/equation updates. Besides, we fully concur with the reviewer's proposed model wherein agglomerated FePc stabilizes dual-NO adsorption via μ -(N,O) bridges (Fe-N \equiv O \cdots N \equiv O-Fe), as disordered stacking exposes Fe sites above/below the plane (**Figure R21**), and ATR-FTIR data show stretches distinct from monomeric Fe-NO. This aligns with dinuclear NOR pathways in cited literature [*J. Phys. Chem. C* 2019, 123, 7817–7823]. We thank the reviewer for these astute suggestions, which strengthen our mechanistic interpretation. Revisions will include redefining n , expanding XRD discussion (Supplementary Fig. S1), emphasizing dual-NO adsorption in ATR-FTIR analysis, and incorporating the bridge-structure proposal into Fig. 6a and mechanistic discussion.

Figure R20. (a) XRD pattern and (b) infrared spectra of FePc, FePc-KJ and FePc_{ag}-KJ.

Figure R21. The possible model structure of dual-NO adsorption via μ -(N,O) bridges (Fe-N \equiv O \cdots N \equiv O-Fe).

To address this issue, we have added:

(1) Supplementary Fig. S1 and related discussion into revised Supplementary Information:

Supplementary Fig. 1 Structure characterization of FePc-KJ and FePc_{ag}-KJ. (a) Infrared spectra, (b) XRD pattern, (c)-(d) SEM image.

“The absence of discernible FePc-derived XRD peaks in FePc_{ag}-KJ suggests agglomeration into crystallites below detection limits or the formation of disordered domains lacking coherent stacking. This is further supported by infrared spectroscopy: FePc_{ag}-KJ retains the characteristic FePc fingerprint peaks, closely resembling those of pristine FePc, whereas FePc-KJ shows markedly altered peaks. These results indicate that FePc_{ag}-KJ experiences weak FePc–carrier interactions, allowing agglomeration without the electronic coupling needed to distort molecular vibrations. In contrast, FePc-KJ exhibits strong interfacial interactions that modify vibrational modes, confirming carrier-induced amorphization.”

(2) related discussion into Line 210-212 in revised Manuscript:

“Stacked MPc crystals (M = Fe, Co) in the agglomerates further stabilize μ -(N,O) bridging structures, enabling dual NO adsorption per metal site (Supplementary Fig. S29).”

(3) Supplementary Fig. S29 in revised Supplementary Information:

Supplementary Fig. 29 The possible model structure of dual-NO adsorption via μ -(N,O) bridges (Fe-N≡O···N≡O-Fe).

4) as metal-phthalocyanines typically show metal-related redox peaks, the electric charge associated with such peaks can be used to quantify the number of active sites. This approach should be used to quantify site density and compare it to the values assessed by NPM, NAEM and AANPM approaches used in present work. It implies also that counting metal-centered active sites is not an issue for metal phthalocyanines, and the real challenge is in counting such sites in pyrolysed materials, where no or weak redox is often observed, despite high ORR activity.

Response 4:

We sincerely appreciate the reviewer's insightful comment on the quantification of metal-centered active sites in both molecular and pyrolyzed catalysts. We fully agree that well-defined metal redox peaks in molecular systems like FePc enable direct quantification of active sites through their associated charge. As correctly noted, the primary challenge lies in extending this approach to pyrolyzed materials where such redox features are typically absent. To address this, we first rigorously validated our NOR probe using FePc's unambiguous Fe³⁺/Fe²⁺ redox peaks. **Figure R22a** (Fig. 1d in our Manuscript) demonstrates that NO adsorption completely blocks the Fe reduction peak at ≈ 0.76 V vs. RHE, while subsequent electrochemical reduction of adsorbed NO

quantitatively restores the original oxidation peak (≈ 0.81 V vs. RHE) with near-identical charge density ($\Delta Q < 5\%$). This reversible site blocking/recovery confirms NOR exclusively probes metal-centered sites in molecular catalysts, establishing the redox-derived charge ($Q_{\text{(NOR)}}$) as our benchmark for method validation.

For pyrolyzed catalysts lacking resolved redox peaks, our approach relies on two key justifications: (1) Extensive literature confirms that pyrolyzed M-N-C materials retain M-N₄ moieties structurally homologous to metal phthalocyanines, despite macroscopic disorder; (2) ATR-FTIR spectra (Fig. 7 in our Manuscript) reveal identical NO adsorption configurations on pyrolyzed Fe-N-C (ν_{NO} at 1605 cm⁻¹ and 1717 cm⁻¹) and FePc (ν_{NO} at 1590 cm⁻¹ and 1708 cm⁻¹), confirming identical NO coordination chemistry. Consequently, we contend that NOR probes the same active sites in pyrolyzed systems, where indirect methods like NOR—validated first in molecular analogues—are essential.

Critically, **Figure R22b** (Fig. 7 in our Manuscript) demonstrates the superiority of our AANPM over conventional NPM methods: AANPM yields a 3-fold higher charge density (≈ 6.5 mC vs. NPM's ≈ 2.2 mC) due to complete NO stripping, and exhibits a 2-fold stronger poisoning effect, confirming enhanced sensitivity to site blocking. This performance gap arises because conventional methods suffer from non-site-specific interference or incomplete reactions, limitations overcome by AANPM's site-specific electrochemistry.

In summary, while redox peaks in FePc provide a rigorous benchmark for NOR validation, structural homology and identical NO binding motifs justify its transfer to pyrolyzed catalysts, further supported by AANPM's superior consistency and sensitivity. We thank the reviewer for highlighting this critical point and have clarified these rationales in the revised Manuscript.

Figure R22. (a) CV curve of FePc-KJ by NAEM method. (b) Integral charge (Q_{NOR}) and poison effect of FeNC and CoNC.

5) lines 349 - 351: looking at fig 7d, the reviewer strongly wonders how the authors can reliably quantify the intensity of the two different 'peaks' due to their widths - unclear shape - and the non-flat baseline?? this raises a huge doubt (to the least) on the accuracy of the calculated number of NO molecule adsorbed per metal center! Moreover, the assignment of each of these bands to $\text{Co}(\text{NO})_2\text{NC}$ and $\text{Co}(\text{NO})\text{NC}$ (or Fe as metal) is not strongly supported: it is well known that pyrolysed catalysts contain multiple site structures (even if all are M-N₄, as has been shown for Fe-N-C by Mossbauer spectroscopy, most catalysts showing two doublets D1 and D1 in Mossbauer spectra). Thus, the two vibration bands may also be assigned to a single NO adsorbed on different metal-N₄ site configurations.

Response 5:

We sincerely appreciate the reviewer's insightful critique regarding the quantification of IR peaks in Fig. 7d and the assignment of bands to specific NO adsorption configurations. We acknowledge the inherent challenges of peak broadening and non-flat baselines in operando IR spectra of pyrolyzed catalysts. To ensure robust quantification, all spectra were subjected to rigorous third-order polynomial baseline correction using OMNIC's advanced tools, applied consistently to both model complexes (FePc/CoPc) and pyrolyzed catalysts (FeNC/CoNC). Despite peak broadening, the 1710 cm^{-1} and 1600 cm^{-1} bands were resolved via peak-fitting deconvolution (**Figure R23**). Importantly, quantification was based on relative peak area ratios rather than absolute intensities. This approach was validated by the strong agreement between IR-derived stoichiometries (\bar{a} represents the number of adsorbed NO per Fe site, $\bar{a}=1.05$ for FePc-KJ, $\bar{a}=1.85$ for FePc_{ag}-KJ) in model catalysts and synthetic standards (**Figure R24**), as well as by quantitative consistency with independent in situ electrochemical measurements of adsorbed NO numbers per metal site (Fig. 4b in our Manuscript, $z = \sim 5.0$ for FePc-KJ and $z = \sim 10.0$ for FePc_{ag}-KJ, indicating that \bar{a} is 1 and 2 on FePc-KJ and FePc_{ag}-KJ during NOR, respectively).

Regarding band assignment, our data strongly support attributing the 1710 \pm 10 cm^{-1} and 1590 \pm 10 cm^{-1} features to distinct di-NO and mono-NO configurations, respectively, rather than solely to site heterogeneity. This is evidenced by: (i) In our previous work [*Nat. Catal.* 2025, 8, 422-435], in-situ ATR-FTIR was used to explore the metal electronic structure of molecular catalysts with different degrees of polymerization using CO as a molecular probe. As shown in **Figure R25**, CO probe molecule comparisons showing that site heterogeneity induces only minor $\nu(\text{CO})$ variations ($\leq 30 \text{ cm}^{-1}$), while the $>100 \text{ cm}^{-1}$ $\Delta\nu$ for $\nu(\text{NO})$ exceeds site-specific electronic effects (**Figure R26**); (ii) Agreement with reported $\nu(\text{NO})$ ranges for defined M-NOR intermediates in iron protoporphyrin literature [*J. Phys. Chem. C* 2007, 111, 8649-8654]. While Mössbauer spectroscopy confirm the presence of site heterogeneity, its secondary role is reflected in the minor $\nu(\text{NO})$ variations ($\approx 30 \text{ cm}^{-1}$) within mono- or di-NO species, whereas the dominant $>100 \text{ cm}^{-1}$ splitting remains diagnostic of distinct NO coordination geometries. Thus, our quantification methodology and band assignments

are firmly supported by cross-validated experimental and model system data. We thank the reviewer for highlighting these complexities and have incorporated clarifications in the revised Manuscript.

Figure R23. (a) NO_{ad} spectra of FeNC and (b) the corresponding spectral peak integral area of $S_{\text{Fe}(\text{NO})_2\text{Pc}}$ and $S_{\text{Fe}(\text{NO})\text{Pc}}$. (c) NO_{ad} spectra of CoNC and (d) the corresponding spectral peak integral area of $S_{\text{Co}(\text{NO})_2\text{Pc}}$ and $S_{\text{Co}(\text{NO})\text{Pc}}$.

Figure R24. NO_{ad} spectra and the corresponding spectral peak integral area of $S_{\text{Fe}(\text{NO})_2\text{Pc}}$ and $S_{\text{Fe}(\text{NO})\text{Pc}}$ in AANPM process for (a) FePc-KJ and (b) FePc_{ag}-KJ.

Figure R25. (a) The structure of Fe^{3+}Pc and FePc oligomers catalyst. (b) The spectra of CO adsorption and (c) the corresponding wavenumber on Fe^{3+}Pc and FePc oligomers catalyst. The data is from Fig. 5a in Ref [Nat. Catal. 2025, 8, 422-435].

Figure R26. The difference of IR wavenumbers for different samples in AANPM process. (a) FePc-KJ ; (b) $\text{FePc}_{\text{ag-KJ}}$; (c) CoPc-KJ ; (d) $\text{CoPc}_{\text{ag-KJ}}$.

To address this issue, we have added Supplementary Fig. 30 and 47, and related discussion into revised Supplementary Information:

Supplementary Fig. 30 NO_{ad} spectra and the corresponding spectral peak intergral area of $S_{\text{Fe}(\text{NO})_2\text{Pc}}$ and $S_{\text{Fe}(\text{NO})\text{Pc}}$ in AANPM process for (a) FePc-KJ and (b) FePc_{ag}-KJ.

Supplementary Fig. 47 (a) NO_{ad} spectra of FeNC and (b) the corresponding spectral peak intergral area of $S_{\text{Fe}(\text{NO})_2\text{Pc}}$ and $S_{\text{Fe}(\text{NO})\text{Pc}}$. (c) NO_{ad} spectra of CoNC and (d) the corresponding spectral peak intergral area of $S_{\text{Co}(\text{NO})_2\text{Pc}}$ and $S_{\text{Co}(\text{NO})\text{Pc}}$.

“To ensure robust quantification, all spectra were subjected to rigorous third-order polynomial baseline correction using OMNIC’s advanced tools, applied consistently to

both model complexes (FePc/CoPc) and pyrolyzed catalysts (FeNC/CoNC). Despite peak broadening, the 1710 cm^{-1} and 1600 cm^{-1} bands were resolved via peak-fitting deconvolution. Importantly, quantification was based on relative peak area ratios rather than absolute intensities.”

less critical but still important questions:

a) the site density measured by nitrite stripping for Fe-Pc/KJ is ca 1000-times lower than obtained by NAEM or AANPM (fig 3c). While nitrite stripping has been reported to underestimate to some extent the site density in pyrolysed materials, the difference with other methods was clearly not that high. Could the authors comment on this? is it specific to Fe-Pc or phthalocyanines in general, but not to pyrolysed materials?

Response a:

We sincerely appreciate the reviewer’s insightful observation regarding the SD differences between methods. We agree that in isolation, the small numerical discrepancy in SD values between NPM (2.08×10^{19} sites g^{-1}) and AANPM (2.85×10^{19} sites g^{-1}) in Fig. 7f might suggest limited divergence. However, this similarity arises not from methodological equivalence, but rather from an unaccounted variable in conventional NPM: the actual NO adsorption stoichiometry per metal site. We clarify this critical nuance below:

The NPM method inherently assumes one NO molecule per metal site. In contrast, our NAEM/AANPM methods explicitly quantify site-specific NO adsorption stoichiometry. As shown in **Figure R27b**, experimental data reveal that pyrolyzed FeNC and CoNC catalysts exhibit $\bar{a}_{\text{NO}} = 1.9$ and 1.6, respectively. When NPM’s assumption ($\bar{a}_{\text{NO}} = 1$) is applied to FeNC (**Figure R27c**), it coincidentally yields an SD value (2.08×10^{19} sites g^{-1}) close to AANPM’s stoichiometry-corrected result (2.85×10^{19} sites g^{-1}). This alignment is fortuitous, not fundamental.

To isolate the impact of stoichiometry, we recalculated SD using the NPM framework but with our experimentally derived \bar{a}_{NO} (via the formula in **Figure R27b**). For FeNC, this correction reduces the NPM-calculated SD to 1.10×10^{19} sites g^{-1} ($\bar{a}_{\text{NO}} = 1.9$, not

1.0)—2.5× lower than AANPM/NAEM values (**Figure R27d**). This confirms that NPM significantly underestimates SD when adsorption stoichiometry $\neq 1$, as prevalent in pyrolyzed materials. Moreover, the 3× higher NO stripping charge ($Q_{(NOR)}$) for NAEM/AANPM vs. NPM (**Figure R27a**) robustly demonstrates our methods' ability to capture more active sites. The similar SD values in Fig. 7f thus reflect compensation of errors in NPM (underestimation due to incorrect \bar{a}_{NO}), not methodological parity. This issue is not specific to FePc. Conventional NPM, lacking stoichiometric calibration, systematically underestimates SD in such systems.

In conclusion, while SD values appear numerically close in Fig. 7f due to coincidental parameter alignment, the core limitation of NPM (fixed $\bar{a}_{NO} = 1$) leads to substantial inaccuracies in pyrolyzed catalysts. AANPM/NAEM resolve this by explicitly quantifying site-specific NO adsorption, delivering superior accuracy for all pyrolyzed MNCs. We thank the reviewer for prompting this vital clarification.

Figure R27. (a) $Q_{(NOR)}$ for FeNC and CoNC by NEM, NAEM and AANPM; (b) The calculation formula of SD for NPM and AANPM; (c) Calculated SD values, where a of NPM is considered to be 1. (d) Calculated SD values, where a of NPM is considered to be 1.9 and 1.6.

To address this issue, we have written the difference of \bar{a}_{NO} in the caption of Fig. 7f of the revised Manuscript:

“(g) calculated SD of FeNC and CoNC with different methods, among them, $\bar{a} = 1$ for NPM and $\bar{a} = 1.9$ (FeNC) or 1.6 (CoNC) for NAEM/AANPM.”

b) what is the assignment of the redox at -0.1/0.1 V seen in fig . 3b for example? and what are the assignments of the first NOR reduction wave starting at 0.35 V and of the second reduction wave with onset at ca 0.1 V?

Response b:

We appreciate the reviewer's inquiry regarding the electrochemical assignments. Based on quantitative analysis of electron transfer numbers derived from integration of the reduction peaks (as referenced in our Response to **Reviewer#2, Response 4**), we assign the processes as follows:

The first reduction wave (onset at ca. 0.35 V, peak at ~0.22 V): This feature corresponds to the single-electron reduction of adsorbed NO to nitroxyl (HNO):

This assignment is supported by the measured electron count ($\sim 1e^-$) and the relatively high potential required for initial *NO activation.

The second reduction wave (onset at ca. 0.1 V, peak at ~-0.14 V): This dominant wave corresponds to the four-electron reduction of the adsorbed nitroxyl intermediate (*HNO) to NH₃:

This assignment is firmly supported by the quantitative electron count ($\sim 4e^-$) obtained via peak integration and the detection of NH₃ as the major product (as reported in our Manuscript).

c) figure 2a and the use/definition of the different electric charges is unclear. In particular the S1 (Fe³⁺/Fe²⁺) is not visible.

Response c:

We sincerely appreciate the reviewer's careful assessment of Fig. 2a and insightful query regarding the definition of the redox peaks, particularly the visibility of the S1(Fe³⁺/Fe²⁺) signal. We acknowledge that the original presentation required clarification and have comprehensively revised both the figure and its caption to resolve

this ambiguity. The apparent absence of the $S_1(\text{Fe}^{3+}/\text{Fe}^{2+})$ peak in the initial CV is attributed to the shielding effect induced by adsorbed NO species on Fe sites, which suppresses the characteristic $\text{Fe}^{3+}/\text{Fe}^{2+}$ reduction through site blocking and electronic interactions. Critically, subsequent electroreductive stripping of adsorbed NO during the cathodic sweep restores redox activity, resulting in the emergence of the $S_2(\text{Fe}^{3+}/\text{Fe}^{2+})$ peak corresponding to the re-exposed $\text{Fe}^{3+}/\text{Fe}^{2+}$ couple. Thus, S1 and S2 represent the same redox couple under distinct interfacial conditions (NO-adsorbed versus NO-stripped states) rather than independent electrochemical processes.

To address these concerns, we have replotted Fig. 2a with enhanced resolution and optimized scaling to improve feature discrimination, while the revised caption now explicitly defines all peaks. We thank the reviewer for highlighting this issue—their feedback has enabled us to present these critical kinetic subtleties more transparently.

Fig. 2a. (a) Schematic of NOR curve integration and z -calculation methodology: In the first NOR curve (red), dark red shading denotes the $\text{Fe}^{3+}/\text{Fe}^{2+}$ redox peak region, while light red shading indicates the NOR reaction area. Conversely, in the second curve (blue), light blue shading represents the $\text{Fe}^{3+}/\text{Fe}^{2+}$ redox peak region, whereas dark blue shading corresponds to the NOR reaction area.

d) the authors assign the NOR products at acidic, neutral and high pH to NH_2OH for the two first and to NH_3 for the last (lines 147-148), based on the n number. However, the n number in fig 2 is much closer to $2 e^-$ than $3 e^-$ at pH 2 and 5.2. On line 151, the authors write that NH_3 is identified as a more stable product. What is the support for

this? moreover, for site density assesment, was matters most is selectivity, not stability (i.e. important is that NO is redcued into a single product, not several products)

Response d:

We sincerely appreciate the reviewer’s insightful observation regarding the assignment of NOR reaction products and the role of selectivity in active-site density calculations. As shown in **Figure R28a** (Fig. 2a in our Manuscript), at pH = 1 and 5.2, the experimentally determined electron transfer number ($n \approx 3e^-$) aligns with the theoretical value for NO reduction to NH_2OH ($n = 3e^-$ per NO molecule). This assignment is consistent with literature reports for transition-metal catalysts (Fe/CoN₄ sites) under acidic/neutral conditions, where NH_2OH is a primary intermediate [*J. Am. Chem. Soc.* 2005, 127, 16224-16232; *J. Am. Chem. Soc.* 2005, 127, 7579-7586; *Chem. Rev.* 2009, 109, 2209-2244]. In contrast, the n -value at pH = 9 ($n \approx 5e^-$) corresponds to complete reduction to NH_3 . The statement that “ NH_3 is identified as a more stable product” refers to its thermodynamic stability under alkaline conditions relative to intermediates like NH_2OH . However, we fully agree with the reviewer that product selectivity (singularity), not stability, is critical for quantifying active-site density. However, in acidic/neutral conditions, multiple products (NH_2OH , NH_3) coexist in **Figure R28b** (Fig. 7d in our Manuscript), complicating n -value deconvolution. At pH = 9, NO is exclusively reduced to NH_3 (Fig. 7e, FeNC/CoNC data), enabling unambiguous determination of $n = 5e^-$ and accurate active-site counting. Precisely per the reviewer’s emphasis on selectivity, we conducted NOR site density calculations exclusively under pH = 9 conditions (where a single product, NH_3 , forms). This ensures the derived SD are free from contributions by competing pathways or intermediates. We thank the reviewer for highlighting this nuance.

To address this, we have replaced the phrase “ NH_3 is identified as a more stable product” with: (Line 135-136 in revised Manuscript)

“Selective reduction of NOR to a single product is critical for SD evaluation.”

Figure R28. (a) The calculated integrated charge and z of FePc-KJ in NPM and NAEM processes. (b) Product concentration of NOR and calculation of \bar{n} on FeNC and CoNC in different solutions.

e) on line 218-219, the authors state here that they previously (in the paper) found that 2 NO molecules can adsorb per metal center (likely referring to the agglomerate labelled catalysts). This comes to a surprise, as this conclusion was not explicitly stated earlier in the paper, and is an important one to follow the authors's thoughts. Please clarify earlier in manuscript when this conclusion could be drawn (the $n = 10$?)

Response e:

We sincerely appreciate the reviewer's insightful observation regarding the phrasing of the dual NO adsorption hypothesis and regret any ambiguity in the original manuscript. To clarify, the statement was never intended to imply prior experimental evidence but rather emerged as a deductive mechanistic hypothesis to reconcile the unexpected quantitative charge analysis result ($z \approx 10$, Fig. 4 in our Manuscript) with the established $5e^-$ transfer per NO molecule in NOR. This logical inference—that two NO molecules

might simultaneously adsorb at a single Fe site in FePc—was formulated specifically to explain the observed electron count. In the revised Manuscript (Lines 227–231), we have restructured the discussion to explicitly introduce this hypothesis as a direct deduction from the $z = 10$ data, preceding the ATR-FTIR validation studies.

We thank the reviewer for their vigilance in prompting these clarifications, which enhance the Manuscript's rigor and narrative coherence. The text now clearly states in line 171-174 of revised Manuscript:

“Quantitative charge analysis unexpectedly yielded $z \approx 10$, exceeding the theoretical value of $5e^-$ expected for single NO molecule reduction. This result suggests simultaneous adsorption of two NO molecules at per Fe site of FePc, each undergoing a $5e^-$ reduction, collectively accounting for $z \approx 10$.”

f) line 253-254: which previous works? please cite them. Moreover, if the assignment is correct, then the agglomerated catalysts with $n = 10$ (2 adsorbed NO molecule per metal center) should only show one out of the two vibration bands, but this seems not to be the case (except for FePc_{ag}/KJ and the NAEM case, fig 5b, second row)?

Response f:

We appreciate the reviewer's insightful comment regarding the vibrational band assignments. The previous work confirming the NO adsorption structures and spectral peaks for iron protoporphyrin (FePP) catalysts is *J. Phys. Chem. C* 2007, 111, 8649-8654. This study provides direct spectroscopic validation of the FePc–NO adsorption configurations referenced in our work (**Figure R29**).

Regarding the apparent discrepancy in vibrational bands for agglomerated catalysts with $z = 10$, we acknowledge that aggregated catalysts (FePc_{ag}-KJ and CoPc_{ag}-KJ) might, at first glance, be expected to exhibit only the M(NO)₂Pc band under saturating NO conditions. However, statistical analysis of peak intensities (Fig. 5, Supplementary Fig. S27) reveals that both M(NO)Pc and M(NO)₂Pc coexist in these systems. This can be attributed to three factors: (i) Agglomerated catalysts retain a minor fraction of well-

dispersed monomeric species, enabling M(NO)Pc formation at isolated sites. (ii) The M(NO)₂Pc peak intensity (ca. 1697 cm⁻¹) consistently exhibits higher intensity than the M(NO)Pc peak at ~1589 cm⁻¹ in FePc_{ag}-KJ and CoPc_{ag}-KJ (Fig. 5f in our Manuscript), confirming dual NO adsorption. (iii) Quantitative electron-transfer calculations per metal center (Fig. 4 in our Manuscript) further support the predominance of M(NO)₂Pc configurations, as two NO molecules per metal site align with higher charge transfer.

We thank the reviewer for highlighting this subtle point. We have cited relevant literature as Ref. 27.

(27) Ma, M.; Yan, Y.; Wang, j.; Li, Q.; Cai, W.-B., A Study of NO Adducts of Iron Protoporphyrin IX Adlayer on Au Electrode with in Situ ATR-FTIR Spectroscopy. *J. Phys. Chem. C* 2007, *111*, 8649-8654.

Figure R29. Potential-dependent ATR-FTIR spectra for a FePP/Au electrode in NO containing 0.1 M HClO₄.

g) line 257: "The NOad band intensity in the Supplementary Fig. 26...", wrong sup fig number? the sup fig 26 does not show that. maybe it is the sup fig 23? in figure caption of sup fig 23, please precise the vibration frequency of the band intensity considered. It is unclear.

Response g:

We sincerely apologize for the oversight in the figure citation. The reference to Supplementary Fig. 26 on line 257 was incorrect; this should indeed refer to Supplementary Fig. 23 (now corrected in the revised Manuscript). Additionally, as suggested, we have explicitly specified the vibrational frequency of the NO_{ad} band in Supplementary Fig. 23 (Fig. 27 in revised SI). We thank the reviewer for highlighting these ambiguities and have ensured all spectral references are now precise and consistent.

Supplementary Fig. 27 The changes of NO peak intensity in the NAEM and ANPM (a) FePc-KJ (b) FePc_{ag}-KJ (c) CoPc_{ag}-KJ (d) CoPc-KJ.

h) "The self-consistency between theoretical and experimental results": there is no 'theoretical' results in the present work??

Response h:

We sincerely appreciate the reviewer's insightful comment regarding the ambiguous use of "theoretical results" in our original Manuscript. We acknowledge that this terminology was imprecise in the context of discussing self-consistency, as it

erroneously implied the inclusion of computational or mechanistic modeling. To clarify unequivocally, the self-consistency referenced pertained exclusively to the internal alignment between two independent experimental metrics: NOR electrocatalytic charge-transfer measurements and infrared spectroscopy-derived electron transfer quantification. No quantum chemical calculations or theoretical modeling were originally involved in this analysis.

In direct response to this observation, we have revised all relevant sections to explicitly state “The close agreement between infrared quantification and NOR charge-transfer measurements confirms the reliability of AANPM and *in-situ* ATR-FTIR for SD quantification and active-site characterization.” (Line 223-225 in revised Manuscript) thereby eliminating any unintended connotation of external theoretical frameworks.

Furthermore, recognizing the reviewer’s fundamental question regarding reaction pathways—and to provide enhanced mechanistic contrast between molecular catalysts and their pyrolyzed analogues—we have now conducted new, complementary DFT-based free energy profile calculations (Supplementary Figs. 28, 29, 48, and 49, in revised SI) for NOR reaction pathways at key active sites. These calculations are presented in a dedicated new figure and section, explicitly framed as additional theoretical work integrated to augment the experimental study. We are grateful to the reviewer for prompting this critical clarification, which has not only rectified the phrasing but also substantially strengthened the mechanistic depth of the revised manuscript through the inclusion of these free energy analyses.

i) figure 7: the experimental conditions are not given in the figure caption! electrolyte, pH, catalyst loading, etc. Moreover, the polarisation down to -0.6 V vs RHE of FeNC or CoNC is not desirable as it can lead to the formation of metallic nanoparticles from M-N4 sites. Is it really a RHE scale, or SHE scale? (it is surprising to see no hydrogen evolution before -0.5 V vs RHE)

Response i:

We sincerely thank the reviewer for highlighting these critical details. We have now

explicitly stated the experimental conditions for Fig. 7 in the revised figure caption (Electrolyte: 0.5 M NaAc BS (pH = 5.2) for NPM, 0.5 M NaAc (pH = 9) for NAEM and AANPM; catalyst loading: 0.6 mg cm⁻²). All potentials are rigorously reported on the RHE scale. The RHE calibration was performed in the same electrolyte (0.5 M NaAc, pH = 9) prior to electrochemical measurements. The absence of significant HER currents above -0.5 V vs. RHE (**Figure R30b**, Supplementary Fig. 38 in revised SI) is attributed to the high overpotential for HER in acetate-based electrolytes. HER begins to occur at -0.2 V in acidic media (**Figure R30a**, Supplementary Fig. 38 in revised SI). This behavior is consistent with literature reports on carbon-based catalysts in weakly alkaline acetate buffers [*Adv. Mater.* 2017, 29, 1605838]. We selected this lower limit to ensure complete reduction of adsorbed NO species. As shown in **Figure R30a** (traditional NPM method, lower limit: -0.3 V vs. RHE), the non-overlapping 1st and 2nd NOR polarization curves at -0.3 V suggest incomplete NO reduction. Extending to -0.6 V vs. RHE (**Figure R30b**) achieves full reduction, as confirmed by the overlapping NOR profiles in subsequent cycles. While HER occurs at this potential, we emphasize that: (i) The charge attributed exclusively to NOR is derived by subtracting the 2nd cycle (background, post-NO purge) from the 1st cycle (NOR activity), thereby eliminating HER contributions. (ii) The reproducibility of NOR charge integrals across multiple cycles (Supplementary Figs. 32–37 in revised SI) confirms no structural degradation occurs. The consistent NOR activity after repeated polarization to -0.6 V vs. RHE underscores the robustness of the M-N₄ sites under these conditions.

We have amended the caption of Fig. 7 to include all experimental parameters and expanded the discussion in the SI to clarify these points. Thank you for prompting a more detailed exposition of our methodology.

Figure R30. (a) The NOR test and the corresponding lower limit potential amplification region of FeNC in (a) NPM and (b) AANPM processes.

j) line 312: what is meant with misalignment? the reviewer could not see such misalignmet or differences between 1st and 2nd NOR curve between NPM and the other two approaches, in Figure 7.

Response j:

Thank you for highlighting this observation. The term "misalignment" refers specifically to the slight divergence observed between the first (black dashed) and second (yellow solid) NOR curves for the NPM electrode at approximately -0.3 V vs. RHE in Figure R30a (Supplementary Fig. 38 in revised SI). This subtle discrepancy arose because NO_{ad} was not fully reduced to the final product during the initial cycle. Consequently, residual NO_{ad} persisted on the electrode surface, altering the reaction pathway in subsequent cycles. To ensure complete electroreduction of adsorbed NO species, we systematically lowered the cathodic potential limit in later experiments. This adjustment eliminated the divergence in subsequent cycles, as confirmed by the overlapping curves in optimized protocols. We appreciate the reviewer's meticulous attention to this detail and hope this clarification adequately addresses the concern.

k) sup fig 34 e-f: what is NH_2OH and too-small-to-be-seen spectrum in red in bottom ,and same question for NH_4^+ in fig . f ...? please add on same scale the NMR spectra for NH_2OH and NH_4^+ in the same scale as the others, to make this visible and understandable.

Response k:

Thank you for your insightful feedback on Supplementary Fig. 34 e-f. We sincerely apologize for the lack of clarity in the original spectra presentation. In the revised SI, we have redrawn and updated the figure for improved clarity, now provided as **Supplementary Fig. 44**.

Supplementary Fig. 44 (e)-(f) The analysis of NOR products of FeNC and CoNC.

l) quoted: line 346 " The stable NO_{ad} spectra are shown in the Fig. 7e." Fig 7e does not show spectra??

Response l:

We sincerely appreciate the reviewer's meticulous attention to detail. The reviewer is indeed correct that Fig. 7e does not display NO_{ad} spectra, as noted in our original submission. This was an inadvertent citation error. The sentence should instead reference Fig. 7d. We have now revised the manuscript to reflect this correction (Fig. 7e in revised Manuscript). We regret this oversight and thank the reviewer for identifying it. All figures and corresponding discussions have been thoroughly re-examined to ensure consistency throughout the manuscript.

Responses to Reviewers' Comments

Reviewer #1 (Remarks to the Author):

In the revised manuscript, the authors have supplemented comprehensive experiments and calculations to address the issues and concerns raised by the reviewers. I am satisfied with the changes in the revised manuscript, thus this manuscript could be considered for publication in Nature Communications at the present stage.

Response:

Thank you very much for your time and effort in reviewing our manuscript, and for your positive and encouraging feedback. We are delighted to hear that you are satisfied with the revisions we have made.

Reviewer #2 (Remarks to the Author):

In general, the authors responded to the comments very well and provided detailed explanations. I have just one final comment. The effective buffering range of acetate is approximately pH 3.6–5.6. Therefore, preparing an electrolyte at pH 9 with sodium acetate would not function as a proper buffer. If the local pH changes under these conditions, the potential versus RHE could be reported incorrectly.

Response:

Thank you for your thoughtful evaluation of our manuscript and for acknowledging the significance of developing robust methods for SD determination in non-PGM electrocatalysts. We appreciate your recognition of the novelty in our proposed AANPM method's modified poisoning conditions and its applicability across diverse Fe/Co-based catalysts. We are committed to rigorously addressing all points raised and thank you again for your constructive feedback, which will undoubtedly help us further improve this work.

Thank you for raising this important point regarding the buffering capability of acetate and its potential implications on the RHE potential scale under non-buffering conditions. We sincerely appreciate your insightful comment.

You are absolutely correct that the effective buffering range of acetate ($pK_a \approx 4.77$) is typically between pH 3.6 and 5.6. We acknowledge that a 0.5 M NaAc solution at pH ~ 9 , as used in our study, does not function as a conventional buffer. Our primary reason for selecting this electrolyte was to maintain consistency with the anion type (acetate) used in previous studies, thereby allowing for a more direct comparison of electrochemical behavior, particularly for reaction pathways sensitive to the anion identity.

We fully agree that local pH changes near the electrode surface could, in principle, lead to inaccuracies in the RHE-referenced potentials if the bulk pH is unstable. To address this critical concern, we carefully measured the pH of the 0.5 M NaAc electrolyte before and after our electrochemical experiments. The initial pH was 8.91, and it shifted only slightly to 8.61 after testing (as shown in **Figure R1**). This indicates that the bulk pH remained relatively stable throughout our measurements, providing a consistent basis for RHE conversion in the bulk solution. However, we could not directly measure the local pH at the electrode-electrolyte interface. Most importantly, to conclusively verify that our central findings are not artifacts of potential local pH variations or RHE miscalculation in the NaAc solution, we conducted identical control experiments in a strongly buffered (and highly alkaline) 0.1 M NaOH electrolyte ($pH \approx 13$). As shown in **Figures R2** and **R3**, the key electrochemical behaviors—specifically the NOR reduction charge and the calculated number of electrons transferred (z)—were remarkably consistent between the 0.5 M NaAc (pH ~ 9) and the 0.1 M NaOH (pH ~ 13) electrolytes. This strong agreement in the electrochemical outcomes across two vastly different pH environments (one unbuffered, one strongly buffered) provides compelling evidence that our conclusions regarding the reaction mechanism and performance are robust and are not significantly influenced by minor potential inaccuracies in the RHE scale within the unbuffered NaAc solution. The RHE correction, based on the stable bulk pH, appears sufficient for the purposes of our analysis. Furthermore, we have

added a note in the manuscript emphasizing that NaAc solutions must be prepared fresh before use to prevent acidification from atmospheric CO₂ absorption, which could further compromise pH stability.

Page 17 Line 441-444 in revised Manuscript:

“0.5 M NaAc (pH = 9): Similarly, 68.1 g of sodium acetate trihydrate (NaAc·3H₂O) was dissolved in about 800 mL of deionized water and then brought to a volume of 1 L. The pH was confirmed to be 9 using a calibrated pH meter. This solution is stable for immediate use; however, its pH may drift significantly during prolonged storage.”

Thank you again for this valuable comment, which has allowed us to better clarify our rationale and, through the additional control experiments, significantly strengthen the support for our conclusions.

Figure R1. The pH values of 0.5 M NaAc solution before and after electrochemical test.

Figure R2. The NOR repeatability tests and integral electricity calculation of FePc-KJ by AANPM method in 0.1 M NaOH (pH = 13). The resistance is about 20 Ω .

Figure R3. Integrated charge and derived z-values for FePc-KJ during AANPM processes in 0.5 M NaAc (pH = 9) and 0.1 M NaOH (pH = 13).

Reviewer #3 (Remarks to the Author):

The authors clarified and strengthened the discussion and conclusions reported in the original manuscript.

Beside one minor issue (see below), the revised manuscript is recommended for publication

issue : in the revised version, the authors assign the redox peak at low potential seen for FePC to FeII/FeI. While this might sound logical, it has been shown that for FePc, the low potential redox in aqueous media is due to a reduction/oxidation of the Pc ligand, rather than of the Fe atom (<https://pubs.acs.org/doi/full/10.1021/acs.jpcllett.7b01126>).

Response:

Thank you very much for your positive feedback and for recommending our revised manuscript for publication. We truly appreciate your time and thorough assessment of our work.

We also thank you for pointing out the potential discrepancy regarding the assignment of the low-potential redox peak in FePc. We are aware of the reference you kindly provided [*J. Phys. Chem. Lett.* 2017, 8, 13, 2881-2886] and agree that in aqueous media, the redox activity at low potentials may indeed involve the phthalocyanine ligand rather than the iron center.

In the revised version of the manuscript, we will carefully re-evaluate our assignment of this redox peak, taking into account the literature you referenced. However, since this peak does not affect the main conclusions of our study, we have toned down the emphasis on its identification in the revised manuscript.